# Modulation of the dynamics of cerebellar Purkinje cells through the interaction of excitatory and inhibitory feedforward pathways

**Yuanhong Tang[1], Lingling An[1]\*, Ye Yuan[1], Qingqi Pei[2], Quan Wang[1], Jian K. Liu[3]\***

**1** School of Computer Science and Technology, Xidian University, Xi'an, China, **2** School of Telecommunication Engineering, Xidian University, Xi'an, China, **3** Centre for Systems Neuroscience, Department of Neuroscience, Psychology and Behaviour, University of Leicester, Leicester, United Kingdom

\* an.lingling@gmail.com (LA); jian.liu@leicester.ac.uk (JKL)

**Data Availability Statement:** The code used to generate the results in this paper is available on https://github.com/jiankliu/Purkinje-Cell-Network.

## Abstract

The dynamics of cerebellar neuronal networks is controlled by the underlying building blocks of neurons and synapses between them. For which, the computation of Purkinje cells (PCs), the only output cells of the cerebellar cortex, is implemented through various types of neural pathways interactively routing excitation and inhibition converged to PCs. Such tuning of excitation and inhibition, coming from the gating of specific pathways as well as short-term plasticity (STP) of the synapses, plays a dominant role in controlling the PC dynamics in terms of firing rate and spike timing. PCs receive cascade feedforward inputs from two major neural pathways: the first one is the feedforward excitatory pathway from granule cells (GCs) to PCs; the second one is the feedforward inhibition pathway from GCs, via molecular layer interneurons (MLIs), to PCs. The GC-PC pathway, together with short-term dynamics of excitatory synapses, has been a focus over past decades, whereas recent experimental evidence shows that MLIs also greatly contribute to controlling PC activity. Therefore, it is expected that the diversity of excitation gated by STP of GC-PC synapses, modulated by strong inhibition from MLI-PC synapses, can promote the computation performed by PCs. However, it remains unclear how these two neural pathways are interacted to modulate PC dynamics. Here using a computational model of PC network installed with these two neural pathways, we addressed this question to investigate the change of PC firing dynamics at the level of single cell and network. We show that the nonlinear characteristics of excitatory STP dynamics can significantly modulate PC spiking dynamics mediated by inhibition. The changes in PC firing rate, firing phase, and temporal spike pattern, are strongly modulated by these two factors in different ways. MLIs mainly contribute to variable delays in the postsynaptic action potentials of PCs while modulated by excitation STP. Notably, the diversity of synchronization and pause response in the PC network is governed not only by the balance of excitation and inhibition, but also by the synaptic STP, depending on input burst patterns. Especially, the pause response shown in the PC network can only emerge with the interaction of both pathways. Together with other recent findings, our results show that the interaction of feedforward pathways of excitation and inhibition,

**Funding:** This work was supported by the National Natural Science Foundation of China (Grants No. 62072355 and 61961130392), Key Research and Development Programs of Shanxi, China (Grant No. 2019ZDLGY13-07), Fundamental Research Funds for the Central Universities and Innovation Fund of Xidian University, China (Grant No. CS2020-10), Zhejiang Lab, China (Grant Nos. 2019KC0AB03 and 2019KC0AD02), and Royal Society Newton Advanced Fellowship, UK (Grant No. NAF-R1-191082). The funders had no role in study design, data collection and analysis, decision to publish, or preparation of the manuscript.

**Competing interests:** The authors have declared that no competing interests exist.

incorporated with synaptic short-term dynamics, can dramatically regulate the PC activities that consequently change the network dynamics of the cerebellar circuit.

## Author summary

It is well known that the dynamics of neuronal networks are controlled by various types of neural pathways that are interactively routing excitation and inhibition converged to post-synaptic neurons. In addition, gating of a specific neural pathway is enhanced by short-term plasticity of the synapses between neurons. However, it remains unclear how a combination of these factors, the strengths of excitation and inhibition, and their short-term dynamics respectively, contributes to the dynamics of single cells and neuronal networks. Using a network model of cerebellar Purkinje cells embedded with the feedforward excitatory pathway from granule cells and feedforward inhibition pathway of molecular layer interneurons. We show that the dynamics of firing rate, firing phase, and temporal spike pattern are notably yet differently modulated by these two pathways. At the single cell level, excitatory short-term plasticity nonlinearly modulates the input-output relationship of firing activity. At the network level, the diversity of synchronization and pause response is governed not only by the balance of excitation and inhibition, but also by synaptic short-term dynamics. Only when both neural pathways are incorporated, there is a strong pause response shown in the network. Our results, together with recent *in vivo* experimental observations in the cerebellum, show that the interaction of feedforward pathways of excitation and inhibition, together with synaptic short-term dynamics, can dramatically change the network dynamics of Purkinje cells.

## Introduction

The dynamics of single neurons and neuronal circuits are controlled by various types of neural pathways involving excitatory and inhibitory neurons. It is generally suggested that the computation in neuronal circuits requires a balanced excitation (E) and inhibition (I) in synaptic transmission through regulation of neuronal excitability [1–5]. In the cerebellum, fine-tuning of E-I balance also plays an important role in cerebellar development and motor coordination [6–8]. Purkinje cells (PCs), as the only output cells of the cerebellum, leverage E-I balance for controlling their dynamics from different neural pathways, involving excitatory inputs from granule cells (GCs) and inhibition inputs from molecular layer interneurons (MLIs) [6, 8, 9].

A sequence of inhibitory inputs through MLIs can rapidly terminate the membrane depolarization of PCs induced by direct excitatory inputs from GCs [10, 11]. In this way, correlated E and I inputs can prevent saturation of the postsynaptic spiking activity and extend its dynamic range for coding of a stimulus. Moreover, developmental changes in the strength of synapses from MLIs to PCs can result in an 11-fold decrease in overall postsynaptic currents [12]. Such a large variation of MLI-PC synaptic strength implies that the MLI activity must be coordinated to efficiently inhibit the activity of PCs. As a result, even a single MLI can dramatically change the PC activity [13].

The neuronal circuit in the cerebellum, as one of the densest networks, employs massive synaptic connections installed with their short-term dynamics for computation [10, 14–16]. One important mechanism for modulating neural excitability is the influence of synaptic transmitter release by synaptic short-term plasticity (STP) at various types of synapses [17–19].

STP can make synaptic efficacy decreased (synaptic depression) or increased (synaptic facilitation) according to inputs from repetitive presynaptic activity [20, 21]. Previous work observed that a wide range of synaptic change was correlated strongly with the strength of GC-PC synapses [22]. As a result, massive mossy fibers from GCs to PCs installed with STP can strongly contribute to the neural dynamics of PCs, which then provide a computational basis for a wide range of behaviors, from motor control to cognition [23, 24].

Therefore, the diversity of excitation gated by the STP of GC-PC synapses, modulated by strong inhibition from MLI-PC synapses, can play an important role in the PC dynamics. Recently, it has been demonstrated that coordinated excitation and inhibition from synaptic short-term dynamics converged to PCs lead to a wide diversity of PC firing dynamics [18, 19]. However, it is still not clear how the interaction of two neural pathways, the large variation of MLI strength in the feedforward inhibitory pathway, together with the STP of excitation raised in GC-PC synapses in the feedforward excitatory pathway, can influence the E-I balance for changing single PC firing dynamics and network behaviors of the PC circuit.

In this work, we addressed this question using a computational model of a neural network consisting of GCs, MLIs, and PCs. Specifically, we incorporated two neural pathways from GCs to PCs. The first one is the feedforward excitatory pathway from GCs to PCs. The second one is the feedforward inhibition pathway from GCs, via MLIs, to PCs. We aim to clarify the specific contribution of excitatory GC-PC synaptic STP and inhibitory MLI-PC strength to downstream PC dynamics. We show that the nonlinear characteristic of excitatory GC-PC STP dynamics can significantly affect PC dynamics in terms of firing rate, firing phase, and temporal spike pattern, which are modulated by MLI inhibition. In particular, excitatory STP enables nonlinear gain modulation. Notably, we demonstrate that the change of synchronization in the network is governed not only by the E-I balance but also by the synaptic STP depending on input burst patterns, whereas the pause response of the network emerges from the tight interaction of two neural pathways. Together with other recent findings, our results show that the interaction of neural pathways of excitation and inhibition dramatically modulate the neural dynamics of PCs that consequently change their network behaviors.

## Methods

### Single cell models

PCs and MLIs were modeled as modified integrate-and-fire neurons [25] that were also used for modelling of cerebellar cells [26]. The membrane potential $V$ obeys the equation:

$$C_m \frac{dV}{dt} = -g_L(V - E_L) - I_{Na} - I_{noise} - g_{AHP}z_{AHP}(t)(V - E_K) - I_{syn}(t) \tag{1}$$

where $C_m$ is membrane capacitance, $g_L$ is leak conductance and $E_L$ is leak resting potential. The sodium current was given by $I_{Na} = -g_L\Delta T \exp[(V - V_T)/\Delta T]$ with $\Delta T = 3$ mV and the firing threshold $V_T$ drawn randomly for each neuron using a Gaussian distribution. When the membrane potential reaches the threshold $V_T$ at the spike $t_{spk}$, $V$ is set to 40 mV for a duration of the spike as $\tau_{dur} = 0.6$ ms. After the spike, at $t = t_{spk} + \tau_{dur}$, repolarizing potential is set to $V_{rest}$, and an afterhyperpolarization (AHP) conductance is activated. The gating variable $z_{AHP}$ follows the dynamics $dz_{AHP}/dt = (1 - z_{AHP})/x_{AHP} - z_{AHP}/\tau_{AHP}$. The resource variable $x_{AHP}$ obeys the dynamics $dx_{AHP}/dt = -x_{AHP}/\tau_{AHP_x} + \delta(t - t_{spike} - \tau_{dur})$, where $\tau_{AHP_x} = 1$ ms. The refractory period is set as $t_{ref} = 2$ ms. To mimic the ongoing activity in our simple point neuron models, a noisy excitatory current $I_{noise} = (V - V_E)g_N$ was injected with a slowly fluctuating conductance $g_N$, described by an Ornstein-Uhlenbeck process,

**Table 1. The parameters of single cell models.**

| Neuron | C(pF) | $g_L$(nS) | $E_L$(mV) | $V_T$(mV) | $V_{rest}$(mV) | $g_{AHP}$(nS) | $E_K$(mV) | $\tau_{AHP}$(ms) |
|---|---|---|---|---|---|---|---|---|
| PC | 250 | 12.5 | -70 | $-50 \pm 1$ | -70 | 4 | -100 | 20 |
| MLI | 20 | 1 | -50 | $-45 \pm 2.25$ | -50 | 4 | -100 | 20 |
| GC | 4.9 | 1.5 | -90 | $-50 \pm 2.5$ | -65 | 1 | -90 | 3 |

$\tau_N dg_N/dt = -g_N + \sigma_N\sqrt{\tau_N}b(t)$, where $\sigma_N = 0.12$ nS, $\tau_N = 1000$ ms, and $b(t)$ is white noise with unit variance density.

Similarly, GCs were modeled as previously based on experimental data [15], whose membrane potential $V$ obeys the equation:

$$C_m \frac{dV}{dt} = -g_l(V - E_L) \exp\left(-(V - E_L)/5\right) - I_{noise} - g_{AHP}z_{AHP}(t)(V - E_K) - I_{syn}(t) \quad (2)$$

where the model components have the similar meanings as the PC and MLI model. All the parameters of neural models of PCs, MLIs, and GCs have the same values, except those listed in Table 1, where the adjustment of parameters for individual cell types were based on previous studies [8, 15, 18, 27].

## Synapse models

For synaptic currents, $I_{syn}$ of MLIs represents the total excitatory input arriving from GCs. $I_{syn}$ of PCs receives excitatory input from GCs and inhibition input from MLIs. $I_{syn}$ of GCs represents excitatory input from mossy fibers (MFs). All the synaptic currents were modeled with a similar form as:

$$I_{syn} = g_{max}r(t)Y(V - E_{syn}) \quad (3)$$

where $E_{syn} = 0$ mV is for excitatory AMPA and NMDA synapses, and $E_{syn} = -80$ mV is for inhibitory GABA synapses. The scaling factor $Y$ is a nonlinear voltage-dependent function for NMDA: $Y = 1/(1 + exp(-(V - 84)/38))$. Otherwise, $Y = 1$ for other types of synaptic receptors.

The gating variable $r$ was described by

$$\begin{aligned} r' &= -r/\tau_{decay} + \alpha.s(1 - r) \\ s' &= -s/\tau_{rise} + Ru\sum_k \delta(t - t_{spk}) \end{aligned} \quad (4)$$

where, $Ru$ is to represent short-term synaptic plasticity with a simple phenomenological model that describes the kinetics of plasticity, such that it treats short-term depression and facilitation as two independent variables, $R$ and $u$, respectively [20, 28].

$$\begin{aligned} R' &= (1 - R)/\tau_{rec} - RU\delta(t - t_n) \\ u' &= (U - u)/\tau_{fac} + U(1 - u)\delta(t - t_n) \end{aligned} \quad (5)$$

Totally, we mainly used four types of synaptic connections between neurons: the excitatory MF-GC, GC-MLI, and GC-PC synapses, and the inhibitory MLI-PC synapse. All types of synapses show a varying heterogeneity of short-term plasticity with mixed fast and slow time scales. When STP is switched off for GC-PC and MLI-PC synapse, the variables $R = 1$ and $u = U$ are held fixed without dynamic updates. Synaptic delays in all synapse are included as 1 ms. In addition, MLI-PC synaptic delays are heterogeneous with a Gaussian distribution (mean as

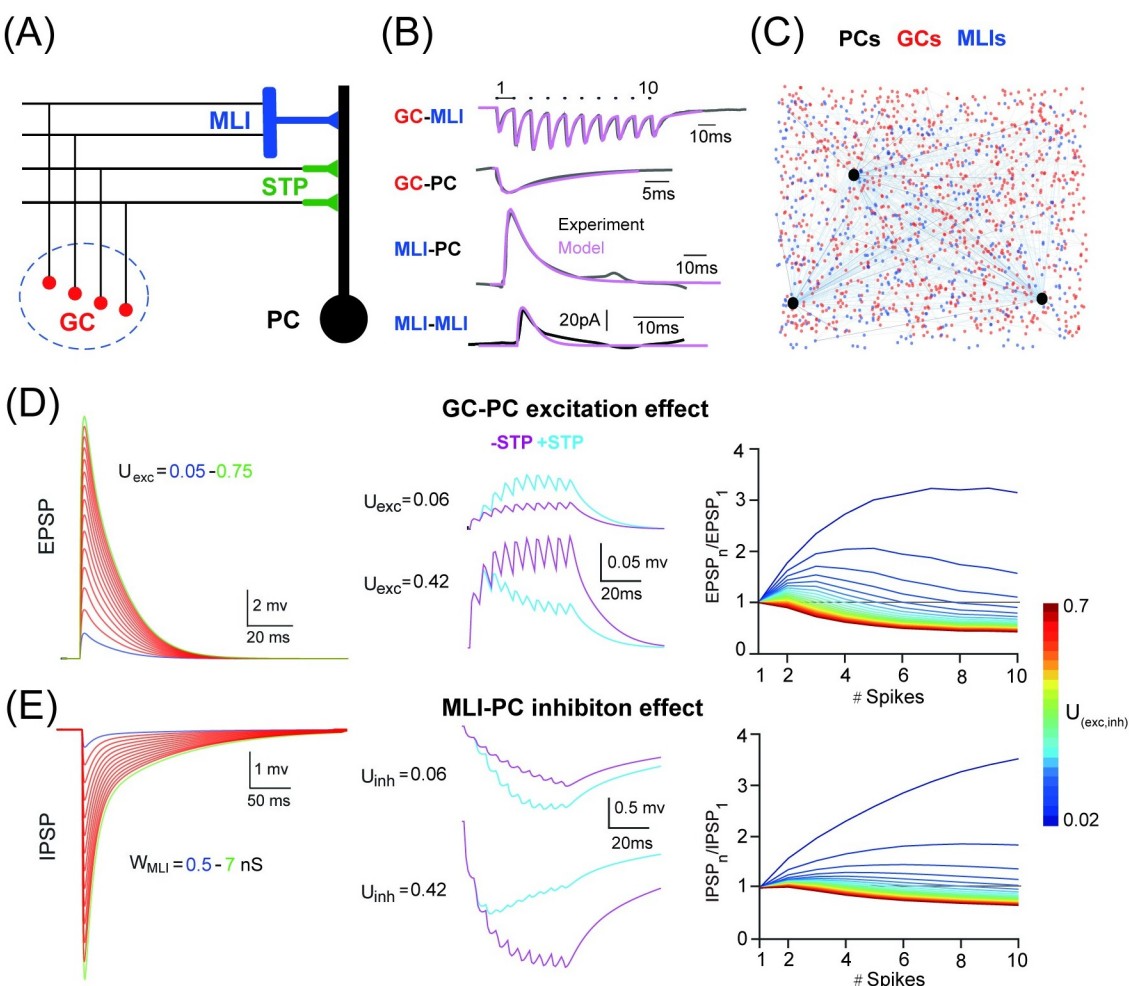

**Fig 1. PC dynamics controlled by excitation and inhibition.** (A) Schematic illustration of feedforward excitatory GC-PC short-term plasticity (STP) pathway and inhibitory GC-MLI-PC pathway on a PC. Granular cells (GCs, red), molecular intermediate neurons (MLIs, blue) and Purkinje cells (PCs, black). (B) Postsynaptic currents of four types of synapses from experimental data fitted by models. (C) The PC network with 50 PCs (black), 1000 GCs (red), and 500 MLIs (blue). For illustration, only 3 PCs are shown. (D) PC in response to the GC-PC input. (Left) EPSPs triggered by a single GC spike by varying GC-PC synaptic STP amplitudes $U_{exc}$ (0.05–0.75 with a 0.05 increment). (Middle) EPSPs triggered by a train of 10 spikes at 200 Hz at two different values of U: $U_{exc} = 0.06$ for facilitation and $U_{exc} = 0.42$ for depression, with (light blue) and without (purple) STP switched on. (Right) STP described by the ratio $EPSP_n/EPSP_1$ showing facilitation or depression in a train of a varying number of burst spikes under different U (0.02-0.7, fixed burst frequency at 200 Hz). (E) Similar to D but for IPSPs triggered by the MLI-PC input. Single IPSPs induced by different strengths $W_{MLI}$ (0.5–7 with a 0.5 increment).

1 ms and SD as 0.2 ms). In case of studying the effect of recurrent inhibition, the MLI-MLI inhibitory synapse was also included. Synaptic parameters for each type of synapse in the model were constrained by experimental measurements as shown in Fig 1B [12, 15, 22, 29]. Postsynaptic currents recorded with current clamps at PCs from experimental data were fitted by models. Specifically, we used the following data: the postsynaptic GC-MLI current under a 10-spike stimulation clamped at -60 mV for MLIs [19]; unitary GC-PC current clamped at -70 mV for PCs [22]; unitary MLI-PC current clamped at 0 mV for PCs [12]; unitary MLI-MLI current clamped at -50 mV for MLIs [30]. The parameters values are described in Table 2. Through the study, two key parameters are focused: the MLI-PC synaptic weight $W_{MLI}$ for varying inhibition strength; the initial efficiency $U_{exc}$ of GC-PC synapses for varying excitation

**Table 2. Synaptic parameters for each synapse in the model.**

| Synapse | | Strength | Synaptic dynamics | | | short-term plasticity | | |
|---|---|---|---|---|---|---|---|---|
| Pre-Post | type | $g_{peak}$(nS) | $\alpha$(1/ms) | $\tau_{rise}$(ms) | $\tau_{decay}$(ms) | U | $\tau_{rec}$(ms) | $\tau_{fac}$(ms) |
| GC-MLI | $AMPA_{fast}$ | 3.2 | 3 | 1 | 1.5 | 0.1 | 100 | 50 |
| | NMDA | 9.6 | 0.35 | 5 | 20 | 0.07 | 50 | 100 |
| GC-PC | $AMPA_{fast}$ | 0.5 | 3 | 1 | 1.5 | $U_{exc}$ | 50 | 400 |
| | $AMPA_{slow}$ | 0.7 | 0.3 | 3 | 8 | $U_{exc}$ | 50 | 400 |
| MLI-PC | $GABAA_{fast}$ | $W_{MLI}$ | 3 | 1 | 10 | 0.1 | 100 | 800 |
| | $GABAA_{slow}$ | $1.5 * W_{MLI}$ | 0.35 | 5 | 100 | 0.05 | 800 | 100 |
| MF-GC | $AMPA_{fast}$ | 1.2 | 3 | 0.3 | 0.8 | 0.5 | 12 | 12 |
| | $AMPA_{slow}$ | 2.4 | 0.3 | 0.5 | 5 | 0.5 | 12 | 12 |
| | NMDA | 2.88 | 0.35 | 8 | 30 | 0.05 | - | - |
| MLI-MLI | $GABAA_{fast}$ | 3.1 | 3 | 1 | 2.5 | 0.5 | - | - |

strength. By default, we set $W_{MLI}$ = 3.5 nS and $U_{exc}$ = 0.4 as guided by experimental data in Fig 1, unless those values mentioned differently in the work below. To further analyze the impact of MLI inhibition on PC network dynamics, we also systematically varied the initial efficiency $U_{inh}$ for MLI-PC synaptic STP.

## PC network model

A network model was set up with 50 PCs, 1000 GCs, 500 MLIs and 500 MFs, where each PC receives synaptic input from 100 GCs and 8 MLIs randomly, each MLI receives input from 4 random GCs. When studying the impact of MLI inhibition on PC network dynamics, we systematically varied two parameters: the number of MLIs targeting a PC for increasing overall inhibition, and the number of MLI-MLI connections for recurrent inhibition before converging to PCs. Therefore, for each PC, there are two input pathways: one direct excitatory input from GC-PC pathway, and another inhibitory input from GC-MLI-PC pathway. The schematic network is illustrated in Fig 1. The spike firing of each GC was generated by injecting a sequence of MF-like spikes as described previously [15]. Each individual GC was activated with different types of spike input patterns, such as Poisson firing, regular firing, modulated firing, and burst firing. Irregular Poisson spike trains were generated based on the method described in [15]. Regular patterns were generated with the same interspike intervals. The input patterns before and after burst are either Poisson or regular spike trains across the population of GCs. The GCs modulation firing inputs were induced by the same probability distribution modulated sinusoidally in time for every trial as in [15]. It is worth noting that each GC was activated with a different onset time, so that all GCs were activated heterogeneously over the time course, except the burst stimulation, where all GCs were activated at the same time. Simulations were run in C++ VS2015 with a time step of 0.1 ms.

## Data analysis

Data collected after simulation was loaded in MATLAB for further analysis. Simulations were run in four major different conditions: the baseline, *i.e.*, PC model running without MLI inhibition and without STP in GC-PC synapses, and three other conditions with MLI on, GC-PC STP on, or both on.

**Gain and offset of the I-O function.** PC firing rates (F) as a function of GC inputs were fitted with the following Hill function as previously [31]:

$$F(GC) = \frac{F_{max}}{1 + (GC_{50}/GC)^n} + F_0 \tag{6}$$

where n is the exponent factor, $F_0$ the firing rate offset, and $F_{max}$ the maximum firing rate. $GC_{50}$ is the value of GC at which F reaches half maximum. To investigate changes in the input-output relationship of PC firing caused by MLI and/or STP, the change of PC response was quantified by $\Delta$Gain calculated as follows [31]:

$$\Delta\text{Gain} = \left(\frac{F'_{+x} - F'_{-x}}{F'_{-x}}\right) \qquad (7)$$

where $F'$ is the average slope of the fits between 5% and 75% its maximum value. +x and -x denote different conditions of with/without STP and/or MLI, *e.g.* ±STP or ±MLI. A shift along the input axis corresponds to an additive operation, while a change in slope corresponds to a multiplicative operation, or gain change. Offset shifts ($\Delta$Offset) were defined as the difference between the half-maximum values of the fits in the conditions +x and -x.

**Spike train analysis.**   To characterize the temporal structure of spike trains, we used three characteristics of spike trains as previously [32]: interspike intervals (ISIs), coefficient of variation (CV) of ISIs, and local regularity $CoV_2 = 2|ISI_{n+1} - ISI_n|/(ISI_{n+1} + ISI_n)$ for a pair (n, n+1) of ISIs.

**Synchronization.**   To characterize network dynamics, the synchronization index for the whole population of PCs was calculated for burst stimuli. Using spike trains of all PCs with a 1 ms time bin, the coefficient of cross-correlation between any two PCs was calculated, *i.e.*, $CC_{ij}$ is the correlation coefficient between PC $i$ and $j$. Then the network synchronization is given by,

$$K_{net} = \frac{2\sum_{i=1}^{N-1}\sum_{j=i+1}^{N} CC_{ij}}{N(N-1)} \qquad (8)$$

where N is the number of the PCs in the network. The average $K_{net}$, based on the firing rate over a time windows, was calculated for a period of post-burst stimulation in different parameters of burst setting. We used 20, 30 and 40 ms for bursts of 2, 5 and 7 spikes at 200 Hz, and 200, 100, 50 and 40 ms for busts of 50, 100, 200 and 300Hz with 10 spikes, respectively.

**Pause response.**   Under the condition of both MLI and GC-PC STP switched on, burst stimulus triggers a prominent pause response after burst in the population spikes of PCs, which is defined by a time interval where there is no spikes in PCs, typically, it is from the off-set of burst to the onset of next wave of population spikes in PC network.

**Phase shift.**   To quantify the phase shift in the output PC spike train relative to the frequency of sinusoidal modulation of GC input spike train, the average PC firing rate over the whole population was fitted by a sinusoidal function $A\sin(2\pi ft + \phi) + C$, where $C$ is the offset of firing rate, and the modulation phase shift at frequency $f$ is thus fully specified by the phase shift $\phi$ of the sinusoidal component, since the initial phase of input stimulus was set as 0 in all simulations.

## Results

### PC dynamics in response to GC input

PC has a unique feature with high-frequency firing activity as the primary information carrier affecting the activity of downstream neurons. The way a neuron transfers information can be represented by its input-output relationship, which is affected by synaptic inputs. Here we examine how inhibitory MLIs and excitatory GC-PC STP are cooperated to affect the PC dynamic sensitivity in response to the GC input.

For each PC, there are two streams of synaptic inputs coming to its dendrites as illustrated in Fig 1A. There is a feedforward inhibitory pathway, the synaptic GC-MLI-PC connection, which consists of GC-MLI synapses with AMPA and NMDA receptors activated by GCs, and MLI-PC synapses with GABA receptors activated by MLIs. Then each PC also receives another feedforward excitatory pathway, the synaptic GC-PC connection consisting of slow and fast AMPA receptors activated by GCs. In addition, there are recurrent connections between MLIs. These four types of synaptic currents can be constrained by experimental measurement shown in Fig 1B (see Methods). To study the network dynamics of PCs, we set up a network with 50 PCs, 1000 GCs, and 500 MLIs illustrated in Fig 1C, where all types of neurons were implemented with modified integrate and fire models and synapses were modeled with short term plasticity (see Methods). The weights of all synapses were drawn from a Gaussian distribution to consider the variability. For each PC, randomly generated network connections were used but with randomized synaptic weights so that the summation of synaptic inputs induces different responses for different PCs (S1 Fig).

When a sequence of mossy fiber spike inputs is delivered, GCs are activated to enable two synaptic pathways onto PCs playing different roles in controlling the firing dynamics. For which, we consider two key features, the excitation of the STP in GC-PC synapses in Fig 1D, and the inhibition strength from MLIs in Fig 1E, as previous evidence show that inhibition from a single MLI can efficiently change PC dynamics [13], and the presynaptic STP can dramatically affect PC activity [18]. We will show that these two factors contribute to PC activity in different ways.

The single spike response of a PC can be gradually changed by the strength of excitation and inhibition, where synaptic strength is regulated by the synaptic weight together with the initial efficacy parameter U of STP. When varying $U_{exc}$ and $W_{MLI}$ systematically, one can change the amplitude of EPSP (excitatory postsynaptic potential) and IPSP (inhibitory postsynaptic potential) recorded at a PC in Fig 1D and 1E). The short-term dynamics of GC-PC and MLI-PC synapses depend on the STP parameters, *e.g.*, the initial efficacy U as in Fig 1D and 1E and time constants, and the input burst frequency (S2 Fig). The parameter U controls the amplitude of synapse release from facilitation to depression in terms of short-term dynamics. With a train of burst spikes at 200 Hz, synapses are facilitating when $U_{exc} < 0.1$ and $U_{inh} < 0.15$ in our model. As the MLI inhibition is more prominent with a large variation of the strength [13], we fixed $U_{inh}$ as 0.1 but changed the weight of MLI-PC synapses to investigate the interaction of inhibition and excitation. However, PC responses vary significantly for different formats of STP dynamics in our study, and the results presented in this work are robust to the change of STP parameters as shown below. We therefore set the default values as $W_{MLI} = 3.5$ nS and $U_{exc} = 0.4$, unless those changed in the following study.

## PC input-output relationship modulated by excitatory STP

In the cerebellum, outside inputs are represented by a sequence of mossy fiber spikes conveying efferent information into a high-dimensional, sparse code of a large population GCs in the network [16, 33, 34]. In the default case, there is no MLI inhibition (MLI off) and no GC-PC STP onto PCs (STP off, *i.e.*, no short term dynamics on GC-PC synapses), while other types of synapses are still dynamic and subjected to the change following the STP rule (See Methods). We aim to examine how these two factors, MLI inhibition, and GC-PC STP, are interacted to change the balance of excitation and inhibition and modulate the dynamics of downstream PCs.

The first fundamental feature of neural computation is the input-output relationship, *i.e.*, I-O function, of PC firing. To investigate this, we used a wide range of input frequencies and

different types of stimulation, which are represented by GC firing activities. Within the context of the I-O function, there are two types of changes in the shape mostly observed for single neurons. The first one is the additive change, where there is a shift of the I-O curve without changing the shape. Another one is the multiplicative change, or gain change, where the I-O function is reduced or increased by a multiplicative factor.

We first leave out MLI inhibition and study the effect of STP of GC-PC synapses by injecting a sequence of Poisson spike trains at different frequencies. Enabling STP can suppress excitatory input conductance and reduce the number of spikes in PCs ([Fig 2A]), which nonlinearly depends on the GC input, such that total excitation $G_{exc}$ is boosted at lower frequencies but suppressed higher input frequencies ([Fig 2B]). Compared to the baseline, adding MLI inhibition produces an approximate additive change of the I-O function with reduced firing. However, using STP can nonlinearly change the I-O function as in [Fig 2C]. Our default values ($U_{exc} = 0.4$ and $U_{inh} = 0.1$) generate depressing excitation on GC-PC synapses and facilitating inhibition on MLI-PC synapses. It has been suggested from experimental data that these synapses could behave in an opposite way [18, 19, 35]. We implemented our model with an opposite pair of values ($U_{exc} = 0.08$ and $U_{inh} = 0.2$) giving facilitating excitation and depressing inhibition, and found the similar results for nonlinear gain control with GC-PC STP ([S3 Fig]). Furthermore, the similar results still hold when the MLI-PC STP was switched off ([S3 Fig]). This confirms the above observation as the effect of STP in GC-PC synapses, rather than the detailed profiles of facilitation or depression in the GC-PC STP. We then characterized the change in slope (ΔGain) and the shift in the half maximal response (ΔOffset) of the I-O relation using fits to Hill functions as previously [31] (see [Methods]). Results in four conditions in [Fig 2D] reveals that GC-PC STP has a greater effect on gain modulation, which is more prominent when both STP and MLI are switched on. Thus, excitatory STP can greatly enhance gain modulation aided by MLI inhibition.

In STP, facilitation needs lower values of U, whereas depression for higher values of U. Thus, fixed U values change scaling of synaptic currents, as well as profiles of STP. Moreover, STP also depends on input frequency, such that at higher input frequencies, dynamical variables of STP, particularly resource variable R saturates with a limited dynamic range to regulate synaptic currents. As a result, differences of gain control is more prominent at lower frequencies. To further characterize the effect of excitation STP and MLI inhibition on the PC I-O function, we systematically varied the strengths of STP $U_{exc}$ and inhibition $W_{MLI}$, respectively. In the absence of STP, there is a limited effect of inhibition when changing $W_{MLI}$ as in [Fig 3A] (top). However, when STP is presented, a sequence of increased inhibition results in a dramatic change of PC firing ([Fig 3A], bottom). Under the same amount of inhibition increment, there is a larger step of gain change with STP than the case without STP.

Similarly, we leave out MLI inhibition while changing $U_{exc}$ by turning STP on and off respectively. Without STP, decreasing excitation strength shows a similar profile as increasing inhibition, as shown by the total excitation $G_{exc}$ in [Fig 3B] (top). The gain change is well displayed, in particular with larger difference for 10-50 Hz GC inputs, whereas less difference at higher frequencies due to saturation of short-term dynamics ([Fig 3B], bottom). This is due to that when STP is present, low synaptic efficacy shows more facilitation whereas high synaptic efficacy shows more depression, which can compensate for the overall change of total excitation (see [S4 Fig] for lower values of $U_{exc}$). These dynamic behaviors are also shown in the profiles of each individual synaptic component of GC-PC synapses ([S5 Fig]). Therefore, these results confirm that GC-PC synaptic STP with the aid of MLI inhibition can significantly enhance gain modulation of PC firing dynamics.

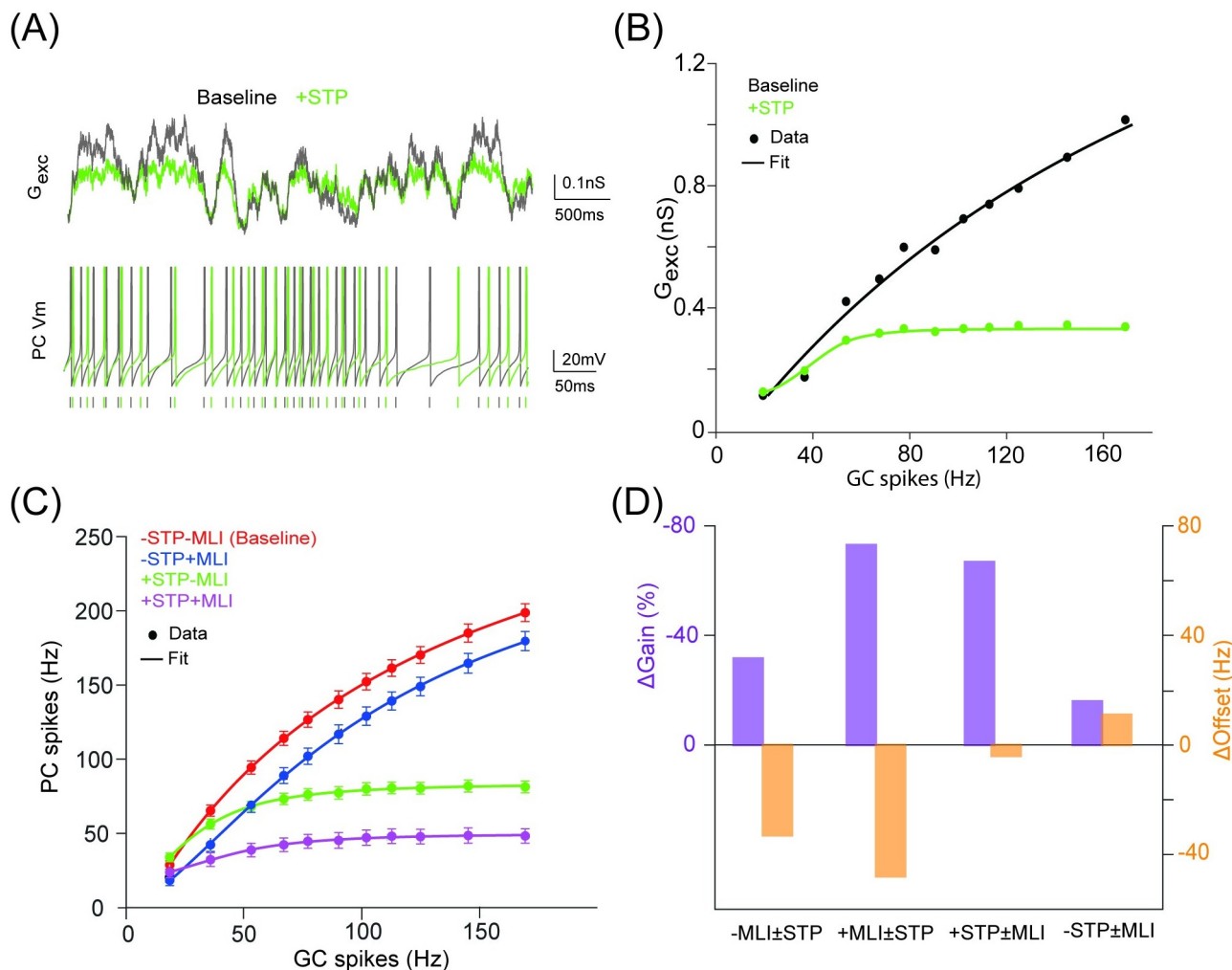

**Fig 2. Inhibition-mediated gain modulation of PC firing dynamics enhanced by excitatory short-term plasticity.** (A) (Top) Sum of excitatory conductance $G_{exc}$ onto a PC in the baseline condition (STP off) and a test condition with STP. The input is 100 independent synaptic trains using Poisson stimulation at 50 Hz. (Bottom) PC membrane potential traces with and without STP. Vertical ticks indicate spike times. (B) Average $G_{exc}$ changing over a range of GC rates with and without STP. $G_{exc}$ was averaged over the time course and all GCs connected to a PC. (C) PC input-output relationship in four conditions, with/without STP and/or MLI. Each point is mean±SD (n = 50). Poisson stimulation was used. Lines in (B) and (C) are fits to a Hill function. (D) Heterogeneous gain and offset changes due to STP (±STD) and MLI (±MLI) from fits in (C).

## PC firing phase modulated by excitatory and inhibitory pathways

Modeling studies suggest that synaptic short-term plasticity can contribute to the phase shift in neural dynamics in response to time-varying inputs at the single cell level [36, 37]. Recent experimental findings in the cerebellum suggest that there is a diversity of phase shifts in different types of cerebellar neurons [38]. Here we investigate how MLI inhibition and STP of GC-PC excitation affect PC responses under oscillating inputs at the population level.

Sinusoidally-modulated inputs with different frequencies were injected into GCs, then drove the PC population firing activity fitted by a sinusoidal curve shown in Fig 4A. When increasing input driving frequency, there is a corresponding increase in the modulated phase shift. Relative phase shifts are easily seen when these fitted sinusoids are overlapped in Fig 4B in the baseline case. Most of the input frequencies generate a phase delay lag of PC firing, except for very low frequency, here 1 Hz modulation. The input with 30 Hz completely

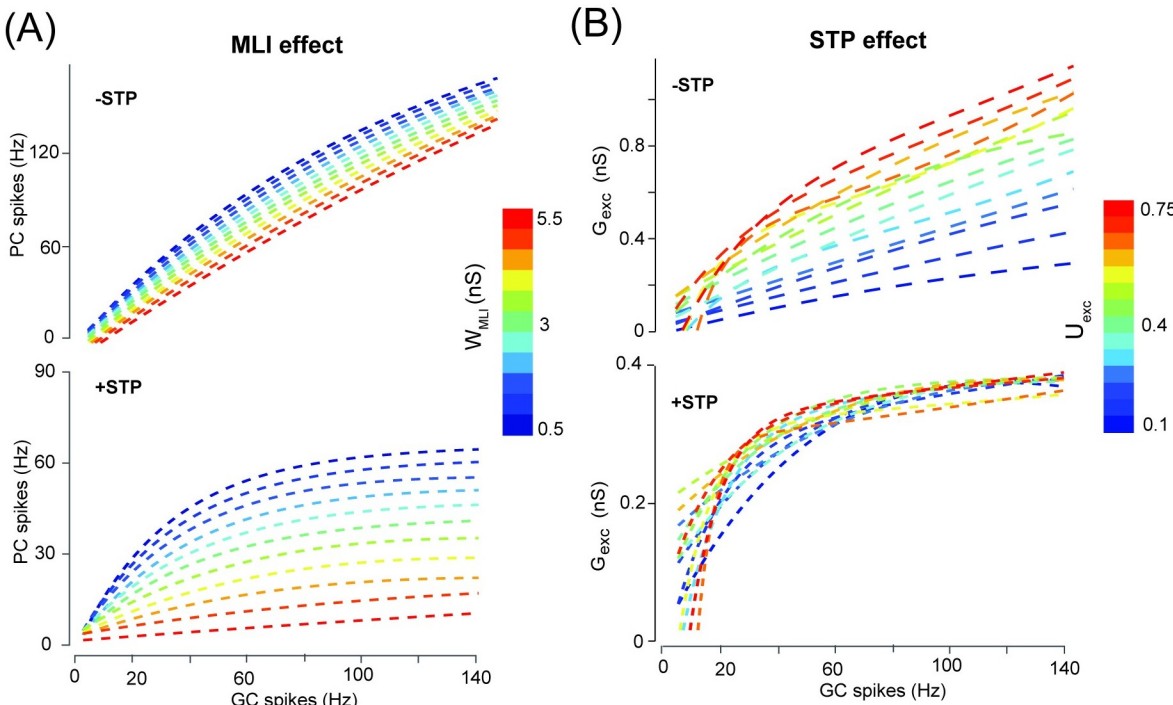

**Fig 3. The gain change of PC firing significantly depending on GC-PC STP.** (A) PC firing modulated by different levels of MLI inhibition strength without STP (top) and with STP (bottom) at $U_{exc}$ = 0.4. (B) Excitatory conductance modulated by different levels of GC-PC synaptic efficacy $U_{exc}$ without STP (top) and with STP (bottom).

reverses the phase in PC firing. Compared to the baseline, phase shifts induced by MLI and STP are different in Fig 4C. MLI inhibitory inputs directly delay PC spike timing and drive large phase delays in PCs, especially, on higher input frequencies. However, excitatory STP results in a relative decrease in the phase shift in response to the same input. Pairing STP with MLI results in a balanced state where the phase shift shows a compromised change as an average of two factors. By varying excitation $U_{exc}$ and inhibition $W_{MLI}$, there is a wide range of systematical phase shifts under a sequence of input frequencies from 0 to 30 Hz in Fig 4D. At the low level of synaptic efficacy as 0.05, strong inhibition inputs trigger few spikes. Otherwise, at other synaptic efficacy values, the increased inhibition dramatically delays phase shift. However, the opposite effect is found for STP of excitation.

As phase shift is related to the time scale of neural dynamics [15], we examined the detailed dynamics with varying time constants of STP. Combinations of facilitation and recovery time constants, ranging from a few to hundreds of milliseconds, enable information transmission over a very wide range of interspike intervals [39]. Fig 5A shows the PC response with different facilitation and recovery time constants at 10 Hz modulation frequency. The change in the phase shift in Fig 5B shows that increasing the recovery time constant results in increased phase shifts at all frequencies. However, increasing the facilitation time constant results in increased phase shifts at a higher modulation frequency (large than 10 Hz), whereas decreased phase shifts at low modulation frequency inputs (less than 10 Hz). Fig 5C shows the systematic change of phase shift for combinations of facilitation and recovery time constants across different input frequencies.

These results suggest that MLI inhibition is sufficiently strong to increase phase delays, whereas STP of GC-PC excitation results in reduced phase delays. Within the STP, the

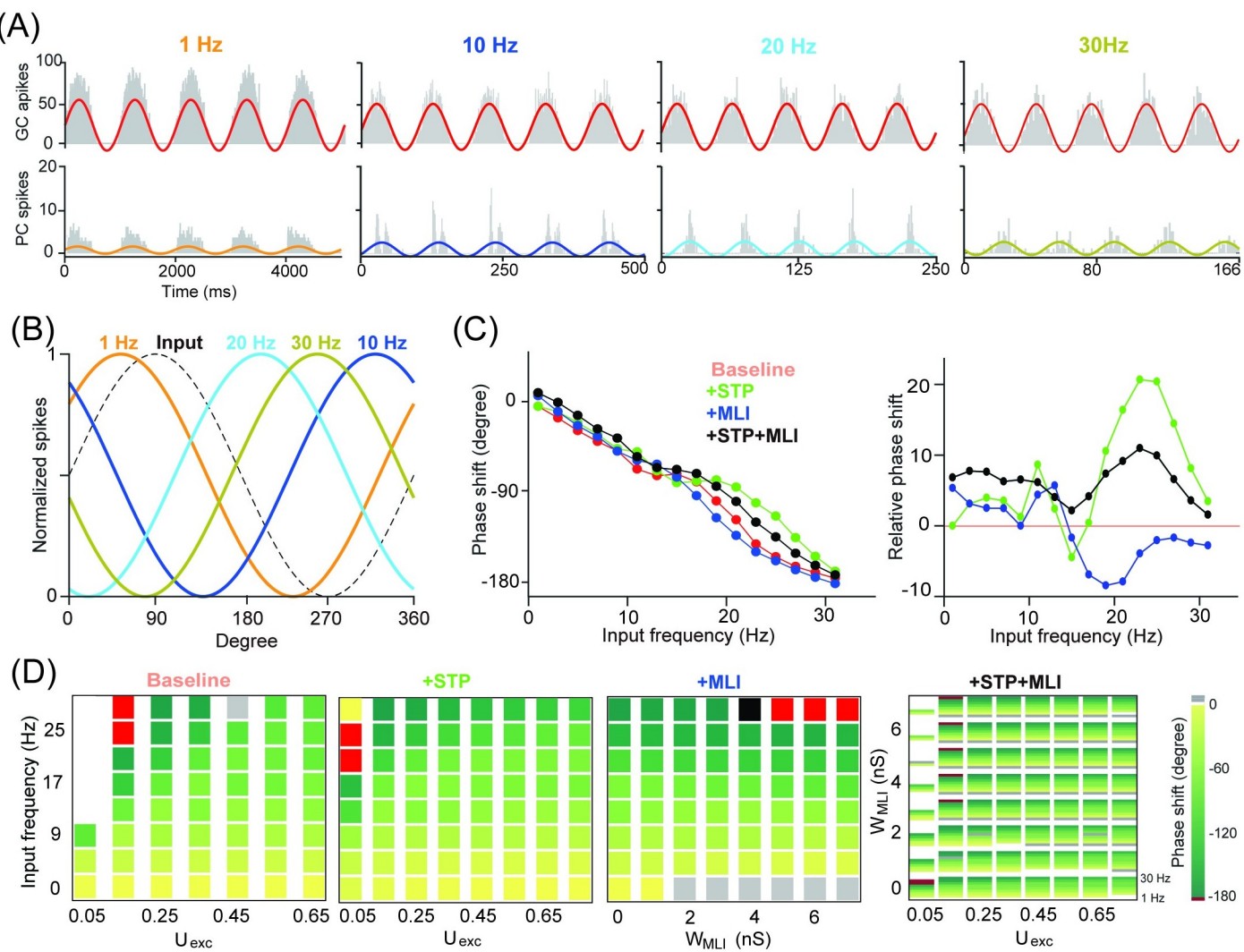

**Fig 4. PC phase modulation affected by MLI and STP.** (A) PC population firing rate (n = 50) in response to GC inputs sinusoidally-modulated with 1, 10, 20 and 30 Hz, from 100 independent input spike trains. Solid color lines are fitted with sinusoidal functions. (B) Normalized fitting curves from (A) show phase shifts relative to the input. (C) Phase shift as a function of input frequency in four different conditions (left), and the corresponding changes of phase shifts relative to the baseline (right). (D) Phase shift changed over a range of excitation $U_{exc}$ and inhibition $W_{MLI}$. (Left) Phase shift over a sequence of input frequencies with different levels of excitation and inhibition in three conditions. (Right) Phase shift changed by combined MLI and STP, where each inner rectangle represents a PC phase shift spectrum over the same sequence of input frequencies (0-30 Hz).

recovery time constant has a greater impact on phase shift than the facilitation time constant. In this way, the balance between excitation and inhibition cooperatively shapes the modulation of PC responses in the network.

## PC temporal spike pattern affected differently by excitation and inhibition

Apart from firing rate, spike temporal pattern has been suggested to play an important role in information processing of neuronal dynamics [40]. Inhibition can control the temporal window of PC discharge and cause highly variable delays in the membrane potential dynamics of neurons [11]. Moreover, STP can induce complex contingencies on the temporal patterns of neural activity [39] and contribute to temporal filtering of synaptic transmission [28]. Here, by

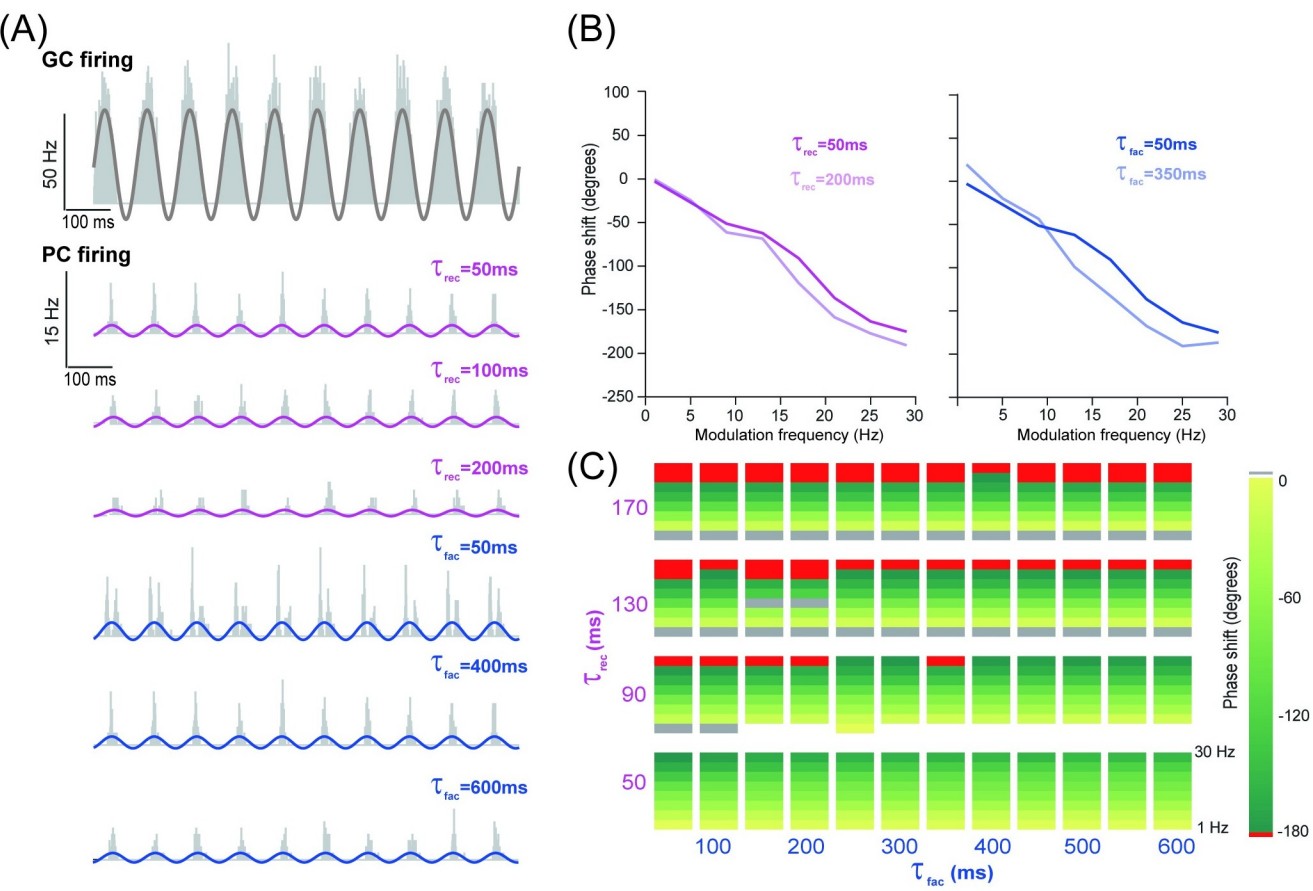

**Fig 5. PC phase modulated by timescales of short-term dynamics of GC-PC synapses.** (A) Histograms of spikes in GCs (top) and PCs (bottom) in response to an input spike train sinusoidally-modulated at 10 Hz, with three examples of time constants of recovery $\tau_{rec}$ (fixed $\tau_{fac}$ = 400 ms) and facilitation $\tau_{fac}$ (fixed $\tau_{rec}$ = 50 ms). (B) Phase shift as a function of input frequency for (left) $\tau_{rec}$ and (right) $\tau_{fac}$, with similar settings as in (A). (C) Phase shift in the parameter space of $\tau_{rec}$ and $\tau_{fac}$. Each inner rectangle represents a PC phase shift spectrum changing over different modulation frequencies (0-30 Hz).

varying inhibition $W_{MLI}$ and excitation $U_{exc}$, we investigate the impact of these factors on the temporal pattern of PC spikes (Fig 6).

As expected, stronger inhibition increases a systematical delay of PC action potentials (Fig 6A and 6B)). However, larger excitation in STP results in contingencies in spike latency such that spike times can be shifted in both ways (Fig 6C and 6D)). Characteristics of the temporal structure of spike patterns, interspike interval (ISI), coefficient of variation (CV) of ISIs and local regularity (CoV$_2$, see Methods), show a larger variation and higher values with both STP and MLI on (Fig 6E). Presumably, these variations could greatly enhance the dynamic range of the PC response. Taken together, these results suggest that the mixture of inhibition and excitation installed with STP can change PC spike patterns in a dynamic coding fashion. Then we will explore this role at the network level.

## PC network dynamics in response to burst input

Finally we investigate how MLI and STP reshape high-frequency bursts of GC activity into PC spike activity. Occurring at a much shorter time scale with a few spikes, a burst at a very high frequency of a few hundred Hz can change the PC response remarkably [41]. Fig 7A shows an illustration of the burst stimulus protocol, where a sequence of 3 or 7 spikes at 100 Hz was

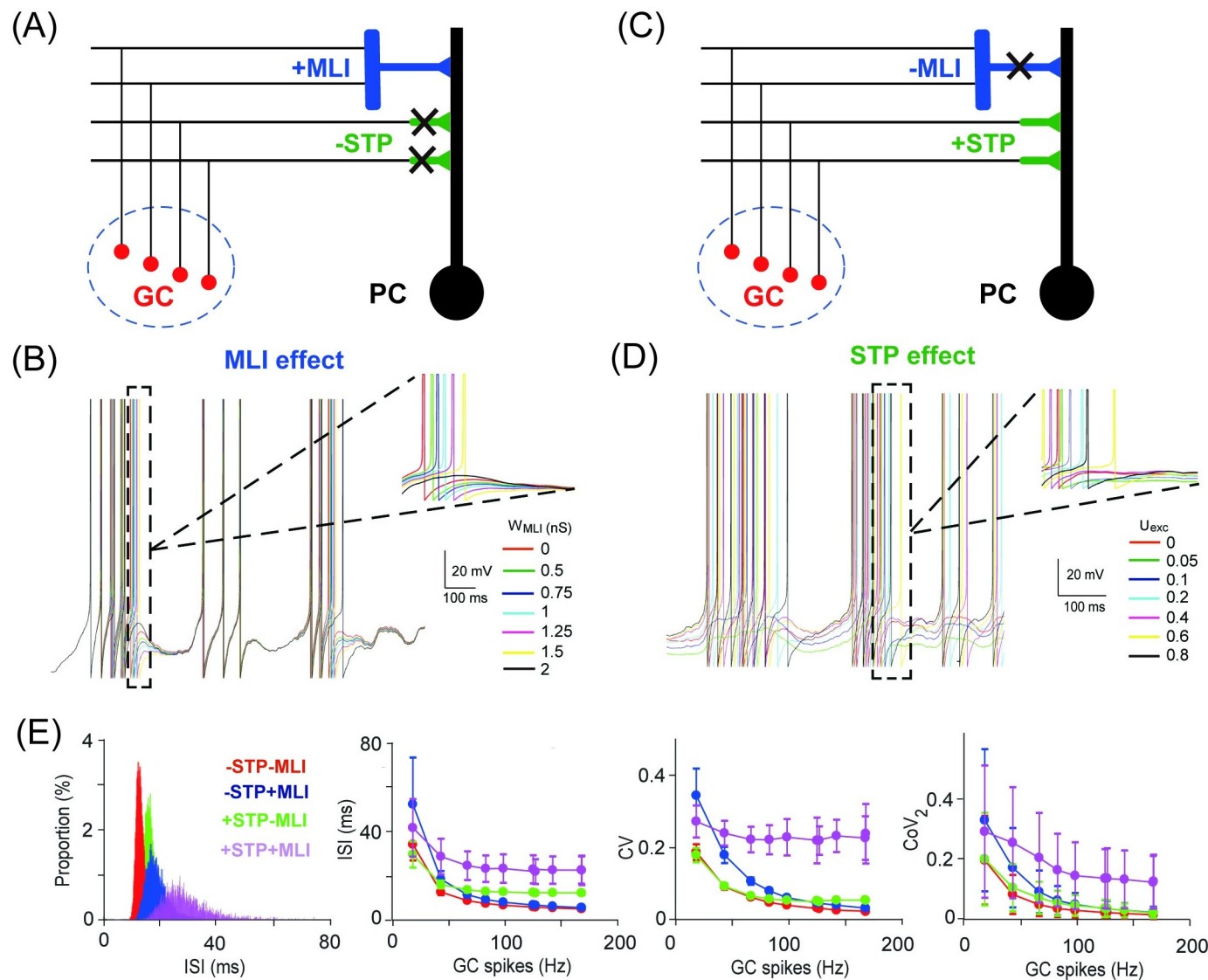

**Fig 6. PC temporal spike patterns affected by MLI and STP.** (A) Schematic illustration of feedforward MLI inhibition and GC excitation when GC-PC STP is off. (B) PC membrane potential traces gradually delayed by different levels of MLI inhibition with an input of 10 Hz Poisson spikes. (C, D) Similar to (A, B) but for MLI off and STP on with varying excitation at different levels of $U_{exc}$. (E) Interspike interval (ISI) distribution of the population PC spikes under four different settings with/without excitatory STP and/or MLI (left). (Right) The change of ISI, CV (coefficient of variation), and $CoV_2$ (local regularity) over a range of GC inputs, averaged over the population of PCs (mean±SD, n = 50).

delivered to each GC, together with the corresponding traces of EPSPs and EPSCs ((excitatory postsynaptic current)) recorded at a PC. Enabling GC-PC synaptic STP has a strong effect of short-term depression under $U_{exc} = 0.4$, compared to the baseline case without STP.

To study the effect of burst input on the PC network, we used a background 20 Hz Poisson spikes first, then a sequence of 5 spikes at 200 Hz was injected into all the GCs in the network to mimic the synchronization of GCs in the granular layer [42] as in Fig 7B. As a consequence, the spike pattern and averaged population firing rate of PCs are triggered. Compared to the baseline without MLI and STP, PC responses are differently tuned by MLI and STP, separately or simultaneously. When both MLI and STP are on, PCs show a prominent response feature: following a burst input, there is a silent period of several tens of milliseconds, named as a

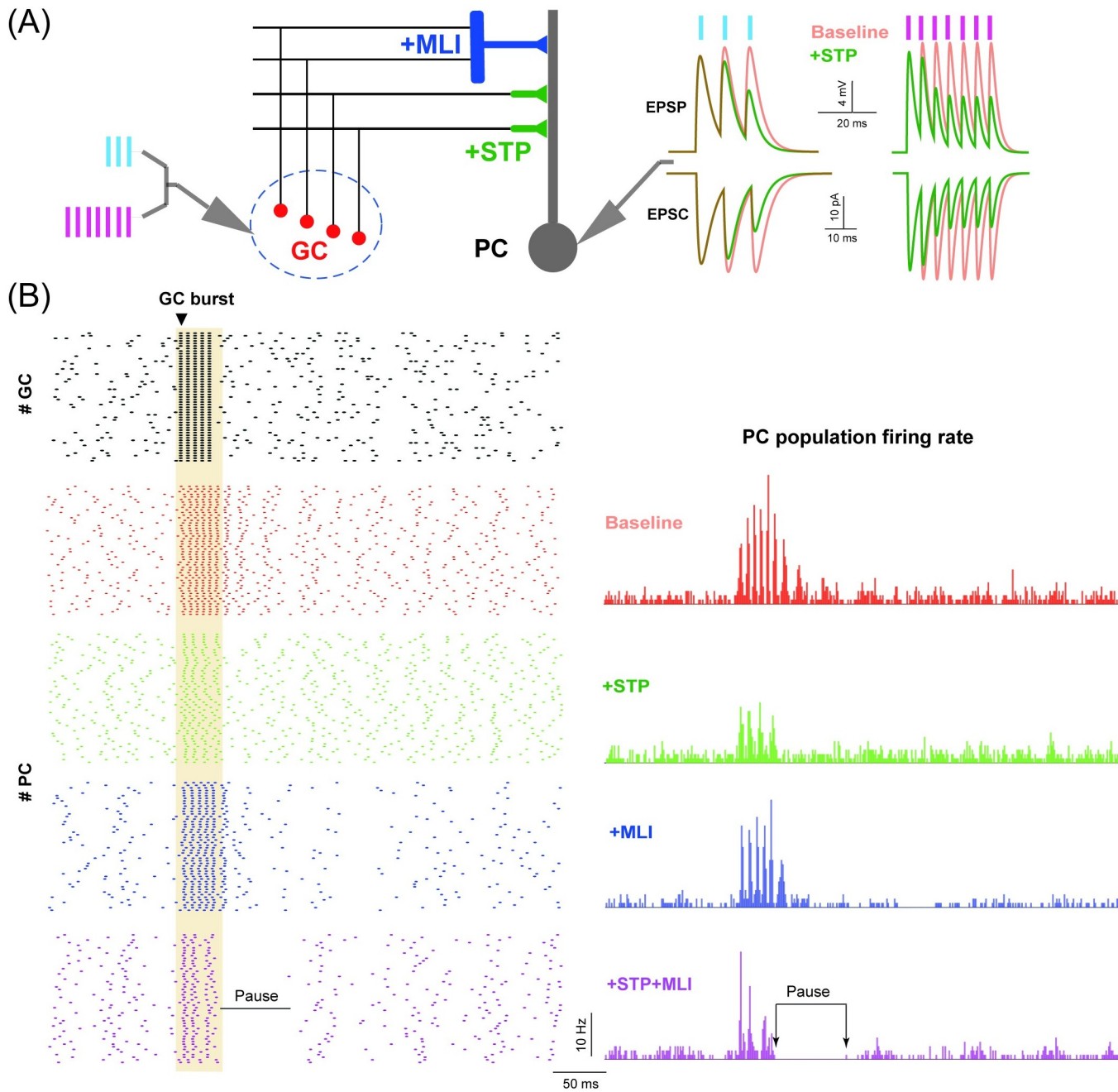

**Fig 7. PC network dynamics in response to burst input.** (A) Schematic illustration of GC-PC and GC-MLI-PC pathways receiving burst inputs. (Left) GCs stimulated by bursts with 3 and 7 spikes at 100 Hz. (Right) EPSPs and EPSCs recorded from PCs for different burst inputs in the baseline condition (red) and with STP on (green). (B) GC spike raster triggered by a burst input of 5 spikes at 200 Hz, only 50 GCs are shown (top). (Bottom) The corresponding spike rasters of 50 PCs (left), and averaged PC population firing profiles in different conditions. Note that the pause response indicated as the time interval between two arrows in the condition of +STP+MLI only.

*pause* characterized by the length of the time interval of PC activity after the burst [43]. Pause response remains when switching off the MLI-PI STP in the model (S6 Fig), and is not depending on the specific choice of GC-PC STP parameters, even using an opposite pair of the synaptic U values that give facilitating GC-PC synapses and depressing MLI-PC synapses

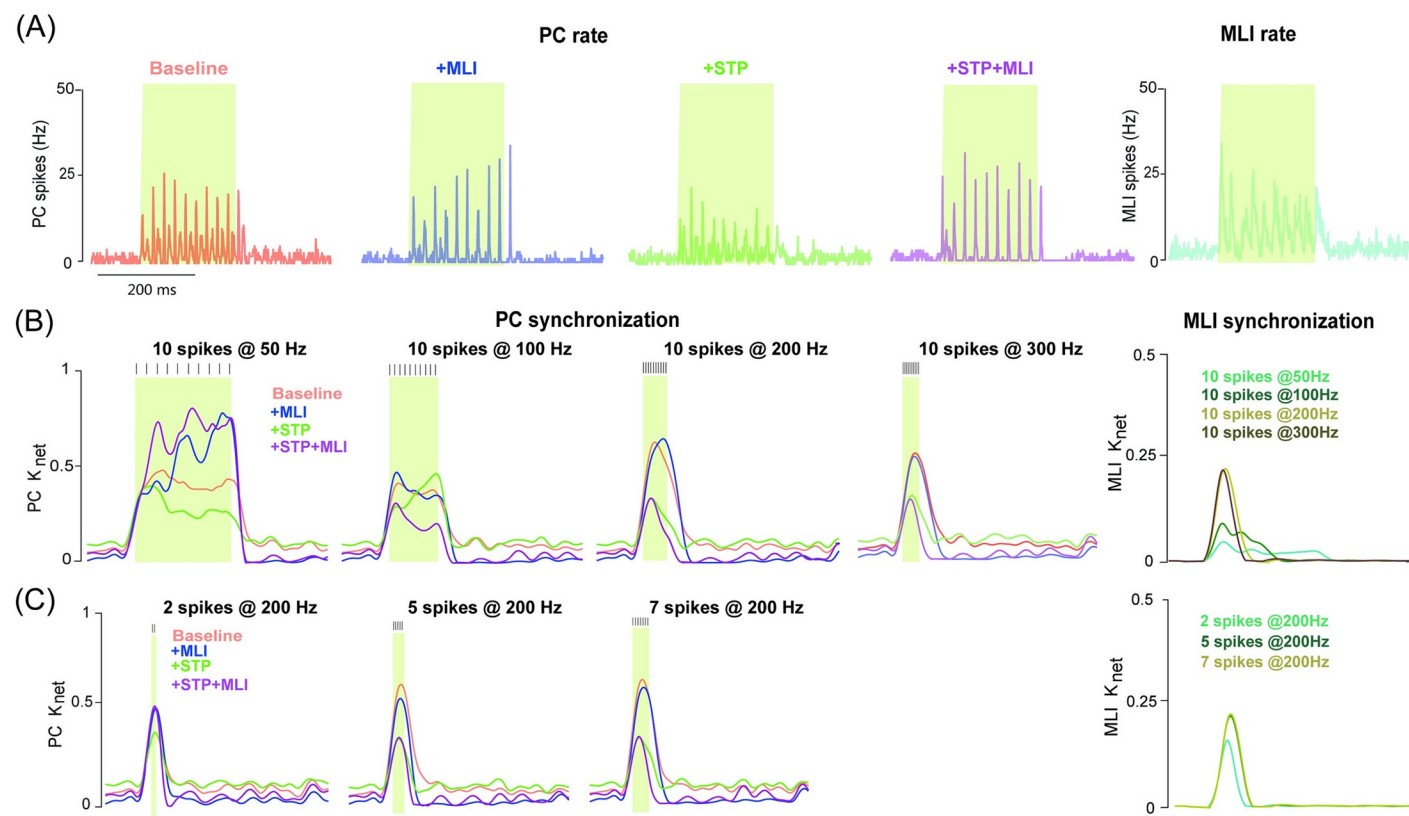

**Fig 8. PC synchronization controlled by excitation and inhibition.** (A) PC population firing rate in different conditions. (Right) The corresponding MLI population firing rate when MLIs are triggered by GCs. The burst input here is 10 spikes at 50Hz. (B) Time course of PC network synchronization computed from the population PC firing rate, in response to input bursts with 10 spikes at 50, 100, 200 and 300 Hz under different conditions. (Right) The corresponding time course of MLI population synchronization. (C) Similar to (B) but for burst inputs at 200 Hz with 2, 5 and 7 spikes, respectively. Colored shadows indicate burst duration. The background noise are Poisson spikes at 20Hz. The time scale bar in (A) applied in all the plots.

(S6 Fig). More detailed analysis shows that during burst inputs, GC-PC synapses become effectively depressed and excitability on PCs is reduced by STP. When MLIs are applied and there is a balanced level of inhibition on PCs, pause can be generated (S7 Fig). These results suggest that the interaction of both neural pathways on PCs, resulting in balanced excitation and inhibition, is necessary to generate pause response.

A unique feature of network dynamics is the synchronization of a neural population. Synchronized PC activity, affecting either accurate timings or rate changes in the downstream nuclei [44], has been suggested to result from the upstream GCs [45]. We conducted experiments on varying the duration of the burst with a fixed number of spikes, to test the burst effect on the PC response. Consistent with the previous observation that input bursts at frequencies up to 300Hz allow PC firing to persist at the network level [41], we found that activation of GCs with 10 stimulus spikes at 50, 100, 200 and 300 Hz, can trigger different types of PC firing dynamics, depending on MLI and/or STP included or not. The PC population firing rate tends to be sustained in the baseline, whereas, MLI builds up dynamics over stimulus, STP generates sharp and transient dynamics, and using MLI and STP together makes the dynamics more transient (Fig 8A and S8 Fig).

To quantify the synchronization of PC firing in response to burst inputs, the index $K_{net}$ was calculated by cross-correlation of the firing rate profiles between a pair of PCs (see Methods). Fig 8B and 8C) shows the temporal change of $K_{net}$ in different conditions with/without MLI

and STP, separately or simultaneously, and under different protocols of burst inputs varying either bust duration in terms of the frequency with a fixed number of spikes, or burst spikes with a fixed frequency. Under burst inputs consisting of a sequence of 10 spikes at 50, 100, 200 and 300 Hz, the change of synchronization shows different behaviors in the network of PCs and MLIs. In general, MLIs show a systematically increasing synchronization with higher frequencies, whereas synchronization of PCs varies according to burst frequency. In contrast, PC synchronization is changed systematically with the increasing number of spikes at a fixed frequency (Fig 8C and S9 Fig), where the PC network was triggered using the different bursts with 2, 5, 7 spikes at 200 Hz. These results suggest that the interaction of inhibitory MLIs and excitatory STP can modulate synchronization in different ways.

To obtain a detailed map of the change in PC network synchronization triggered by bursts within the parameter space of inhibitory MLI $W_{MLI}$ and excitatory STP $U_{exc}$, we quantified the gain of synchronization defined as the relative change calculated by the averaged $K_{net}$ of test conditions (with/without MLI and STP) subtracted by that of the baseline. Fig 9A shows that there is a large variety of changes across conditions. Generally speaking, weaker inhibition results in decreased synchronization, except the case for the burst with 2 spikes at 200Hz, where spikes are too few to obtain the significant change. Similarly, we computed the pause response as in Fig 9B. It is clear that the change of pause is more systematic. The network develops a systematic pause with longer duration after burst input with increased MLI inhibition, which is expected due to that inhibition prolongs the pause period. In addition, low excitatory synaptic efficacy tends to enhance pause duration when inhibition strength is fixed, since the excitation is too weak at low values of $U_{exc}$ and inhibition is more stronger.

So far we varied excitation and inhibition using excitatory STP strength $U_{exc}$ and inhibitory MLI strength $W_{MLI}$. Total excitation and inhibition are contributed by a set of four parameters, weights of GC-PC and MLI-PC synapses and their STP strengths. We then systematically vary the parameters of STP in both $U_{exc}$ excitatory and $U_{inh}$ inhibitory synapses. In contrast to the nonlinear behaviors shown above, the paired change of U values results in a linear behavior of the gain of synchronization (Fig 10A). Such that excitation and inhibition have an opposite effect on the gain: higher excitation reduces gain, whereas higher inhibition increases it. However, the U strength of excitatory plays a dominant role in controlling of pause response as shown in Fig 10B, where there is little effect of $U_{inh}$. Therefore, a large variation of MLI inhibition strength, rather than short-term dynamics of MLI-PC synapse, is essential to the nonlinear tuning of PC network dynamics.

The results above indicate that it is MLI inhibition that plays an important role in PC network dynamics. We then systematically vary inhibition strength by increasing the number of MLI-PC synapses per PC from 1 to 8 MLIs (Fig 11A). Consistent with previous results, stronger inhibition generates a large change for PC synchronization. Depending on the burst protocol, there is more or less synchronization changing over MLI inhibition. However, the pause response is systematically large with stronger inhibition. Another feature for MLIs is that there are recurrent connections between MLIs [30]. We then generate different recurrent connections between MLIs by increasing the number of MLIs-MLIs synapses per MLI, while fixing the inhibition level using 8 MLIs per PC (Fig 11B). There is no systematic change across different levels of recurrent MLI connection. Thus, it is the inhibition strength from MLIs that determines PC synchronization and pause response.

Taken together, these findings suggest that both synchronization and pause response in the PC network are affected by MLI inhibition and GC-PC synaptic STP with the transition from facilitation to depression. However, the major role in controlling pause response is MLI inhibition, where short-term dynamics of MI-PC synapses and recurrence of MLIs less affect the pause response of PC network dynamics.

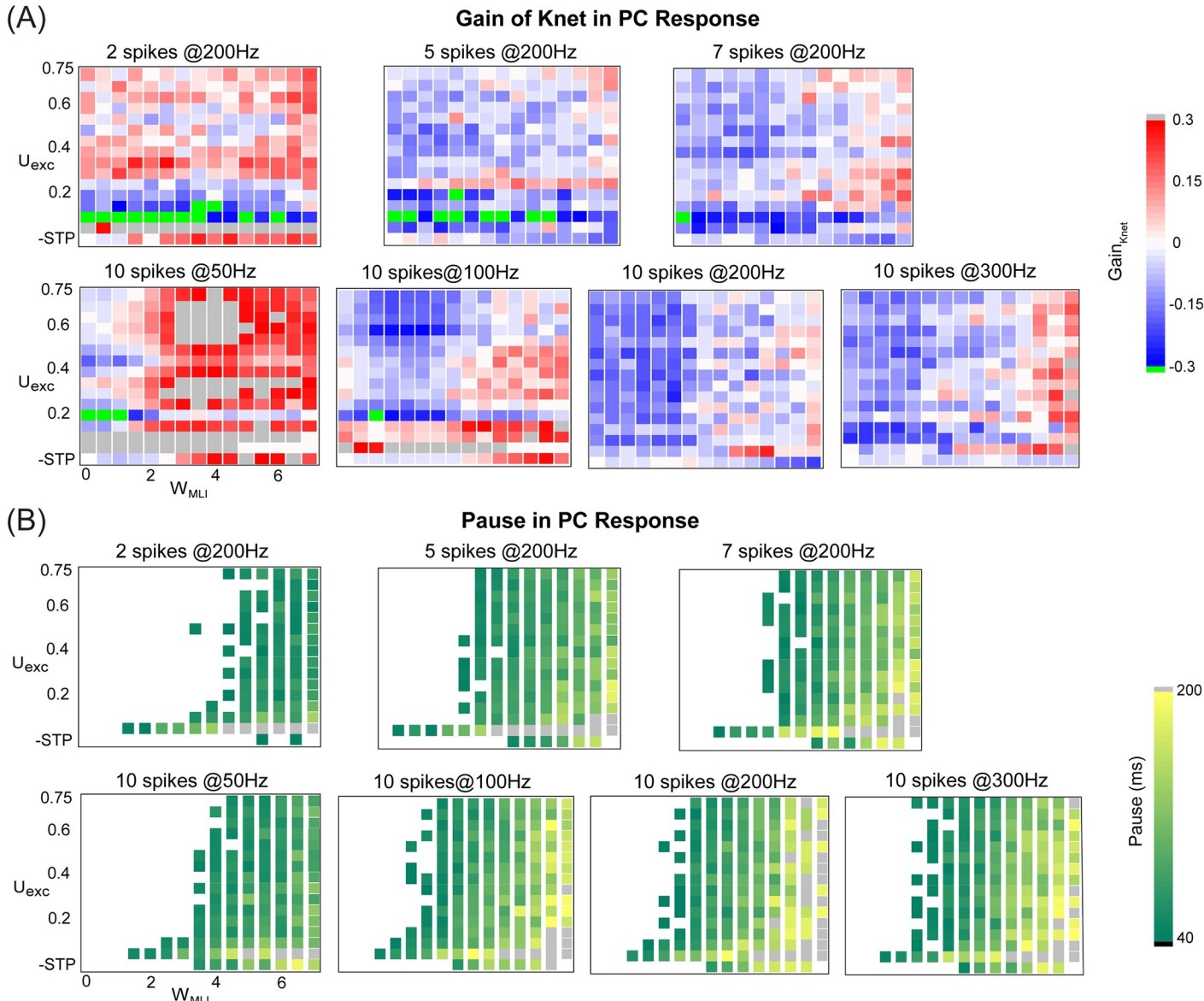

**Fig 9. The change of PC synchronization and pause response in the parameter space of excitation and inhibition.** (A) The gain of synchronization changed in different ways with $W_{MLI}$ and $U_{exc}$ under bursts with different spikes and frequencies. Each inner rectangle in each panel represents a single gain value of synchronization under one burst protocol. (Top) Bursts with 2, 5 and 7 spikes at 200 Hz. (Bottom) Bursts with 10 spikes at 50, 100, 200 and 300 Hz. In all plots, note that there is no STP but static excitation with $U_{exc} = 0.4$ in the last row of the matrix. Similarly, there is no MLI inhibition in the first column of the matrix. Thus the first point at the left-bottom corner of the matrix is the baseline. The gain was defined as the relative change calculated by the average $K_{net}$ compared to that in the baseline. (B) Similar to (A) but for pause response induced by bursts.

## Discussion

In this study, using a network consisting of Purkinje cells with their upstream excitatory granular cells and inhibitory molecular layer interneurons under different stimulation protocols of mossy fiber spiking inputs, we explored how the spiking dynamics of PCs are modulated by two neural pathways from GCs to PCs: direct excitatory GC-PC pathway and feedforward inhibitory GC-MLI-PC pathway. We focused on GC-PC synaptic short-term plasticity and

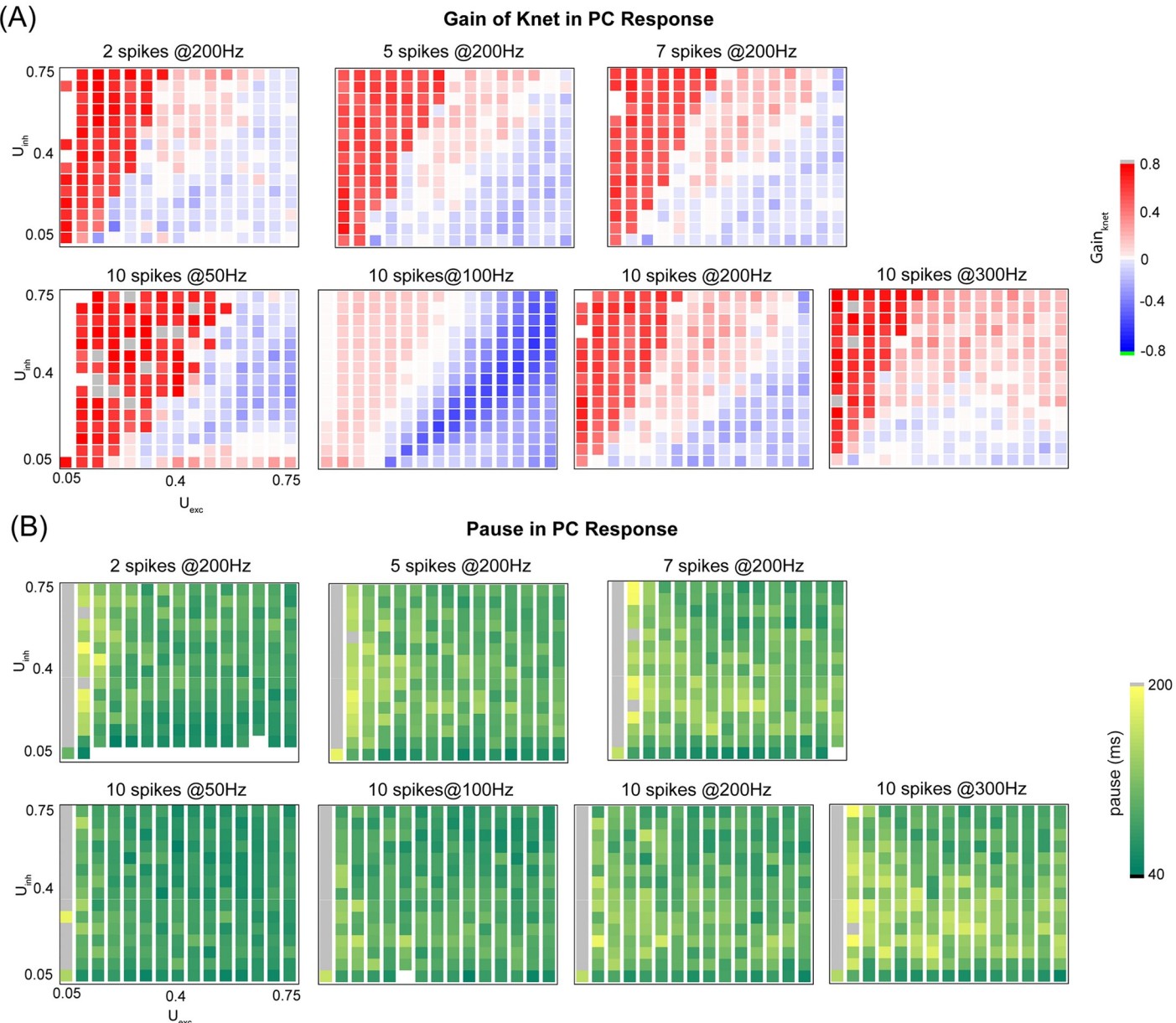

**Fig 10. PC synchronization is controlled by STP of both excitation and inhibition, whereas PC pause response is less dependent on STP of inhibition.** (A) The gain of synchronization regulated differently by STP of ML-PC inhibition $U_{inh}$ and excitation GC-PC $U_{exc}$ under bursts with different spikes and frequencies. Each inner rectangle in each panel represents the gain of network synchronization under one burst protocol. (Top) Bursts with 2, 5 and 7 spikes at fixed 200 Hz. (Bottom) Bursts with 10 spikes at 50, 100, 200 and 300 Hz. Note that the gain was defined as the relative change calculated by the average $K_{net}$ with STP subtracted by that without STP. (B) Similar to (A) but for pause duration induced by bursts.

strong MLI-PC inhibition to regulate PC spiking activity at the level of single cells and networks. On the single-cell level, the input-output firing dynamics, temporal spiking pattern, and phase shift, are nonlinearly modulated by these two factors. On the network level, the synchronization and pause response are modulated by excitation and inhibition. Notably, nonlinear gain control is achieved by excitatory STP, and pause response is controlled by the interaction of both neural pathways.

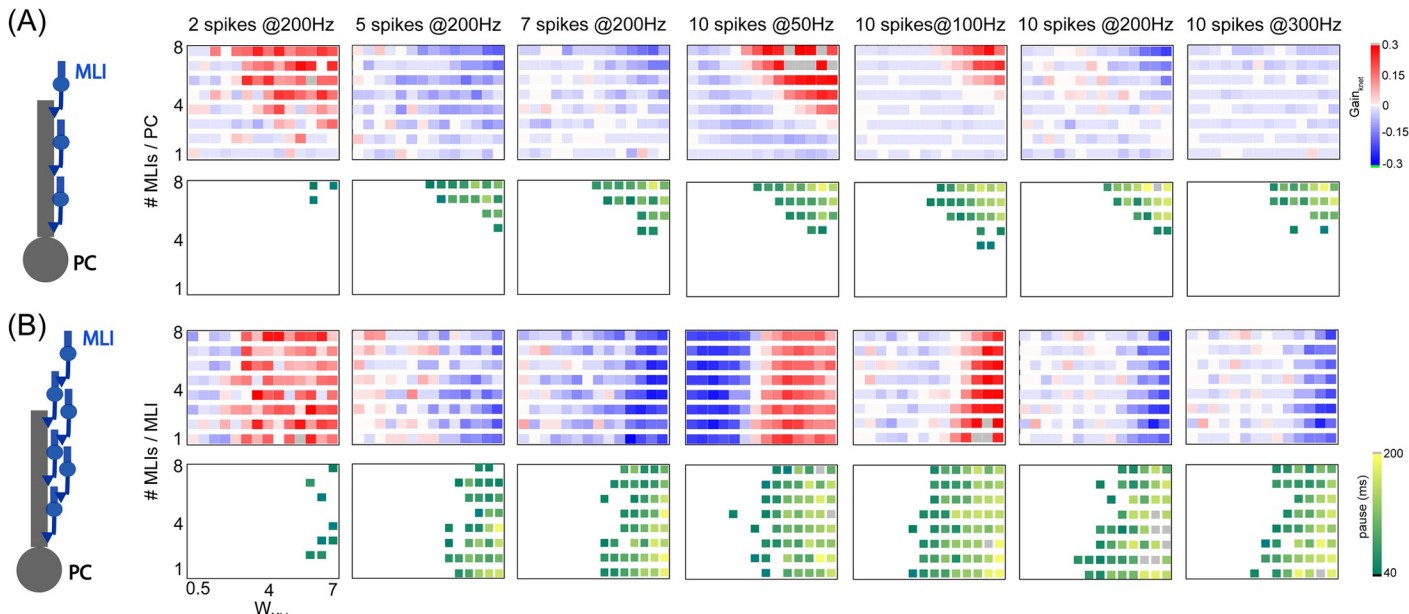

**Fig 11. PC synchronization and pause response controlled by MLI inhibition.** (A) PC network with no recurrent MLI inhibition. (Left) Illustration of the PC network receiving different numbers of MLIs. PC synchronization (top) and pause response (bottom) affected by MLI inhibition with stronger weight or more number of MLI-PC synaptic connections per PC, under burst inputs with different spikes and frequencies. Inner rectangles in each panel represent the gain of synchronization under one burst protocol. The gain of synchronization was defined as the relative change of the average $K_{net}$. (B) Similar to (A) but for the PC network with recurrent MLIs included, such that each MLI receives 1-8 MLIs recurrently. Here in all plots, each PC receives 8 MLIs.

## Inhibition-mediated PC dynamics modulated by short-term plasticity of excitation

Inhibition is a unique feature for shaping neural dynamics in neuronal circuits [46]. In the cerebellum, the role of MLIs is significant in shaping PC dynamics [13, 14], with strong synaptic strength changing up to 11 folds for postsynaptic PCs [12]. PCs receive direct MLI inhibitory inputs from stellate cells at pericellular terminals known as spiny dendrites, and also from basket cells that terminate on the smooth shafts of dendrites [47]. A large variety of inhibitory strength depends on the specific location rather than specific types of cells. Although there are various types of short-term dynamics in MLI-PC synapses [18, 19], the determinate role in controlling PC dynamics is more dramatic when turning on or off a single MLI, which can alter downstream PCs significantly shown by *in vivo* experiments [13]. MLIs also show spontaneous firing activities in the absence of excitatory inputs [48, 49], which make inhibition more prominent to PCs. Thus, it is expected that the effect of the MLI inhibition strength is more significant in the PC network [13].

On the other hand, the excitation to PCs is controlled by the short-term dynamics of GC-PC synapses [18, 22]. There is a large variation of synaptic modifications due to different release probabilities of GCs, such that short-term plasticity of GC-PC synapses can lead to a wide range of changes in release probability [22]. Therefore, here we stressed MLI inhibition and GC-PC synaptic STP together and studied how short-term plasticity of excitation changes the dynamics of MLI-medicated PCs. MLIs are highly sensitive to excitatory inputs [29], such that increased weight of inhibition can narrow the time window for the integration of excitatory inputs. As a result, different contributions of MLI and GC-PC STP could affect the time window, especially changing the precise synchronous firing time of PCs.

## Input-output function of PC firing

How neurons respond to stimuli is one of the central questions in neural coding. The stimulus-response, or input-output, relationship of single neurons is a fundamental feature for neural dynamics. A change in the gradient input-output function of a neuron, termed gain modulation, is associated with several factors, including shunting inhibition, synaptic noise, and dendritic morphology [50–53]. Here, we investigated the contribution of synaptic input to gain modulation taking into account the effect of MLI inhibition and STP of excitation, which can modulate PC discharges in response to GC inputs. Our findings show that the effect of inhibition is an approximate additive change of gain modulation, which is related to the fast onset of feedforward inhibition that can rapidly limit excitatory postsynaptic potentials to reduce responsiveness to inputs [10, 11]. In contrast, excitatory STP enhances gain changes due to highly nonlinear synaptic behavior. However, short-term depression and facilitation modulate gain changes differently, since short-term depression is functionally like low-pass filtering, whereas short-term facilitation acts as high-pass filtering. Thus, the way of gain modulation can be altered by controlling of STP even with the same input.

The functional role of STP in nonlinear gain control is generally demonstrated [54] and also observed in GC recordings with dynamics-clamp inputs mimicking synaptic spiking inputs [31]. Consistent with these observations, we found nonlinear gain control exists in a wide range of model settings parameters of short-term dynamics of synapses. Linear PC I-O function could be a specific case of nonlinear gain control under certain conditions [55]. Our results suggest that the diversity of nonlinear gain control in different degrees could enhance neuronal computation using different coding strategies and make the cerebellar microcircuit as an adaptive filer [56].

## Phase modulation of PC firing

The impact of synaptic dynamics on the phase of neuronal responses is potentially significant in a wide range of neuronal systems, particularly for sensory processing and generating motor outputs in the cerebellum [15, 38, 57]. Here we found that PC phase modulation at the population level can be determined by the input frequency, but can also be adjusted by inhibitory MLI and excitatory STP differently.

Phase modulation by inhibition is sensitive to the input frequency and exhibits in-phase at low frequency and phase lag at high frequency, especially for higher inhibition intensity, which is due to that inhibition can delay spike timing. In contrast, phase modulation by excitatory STP exhibits phase lead at high frequency. As STP can show both depression and facilitation, depressing synapses trigger spikes after a long period of presynaptic inactivity, whereas facilitation synapses are most effective at transmitting at the end of a burst of activity [39]. Our results indicate that combined inhibition and excitatory STP act on spike times in a nonlinear way, which then extends the PC encoding capability to modulate the phase to downstream neurons.

## Synchronization of PC firing

For neural temporal codes involving spike pattern and temporal structure in spike trains, previous studies suggest that PC synchronization has a great effect on the computation carried out by downstream neurons [27, 44, 45, 58, 59]. There are several proposed mechanisms contributing to PC synchronization, including GC inputs, ephaptic coupling between PCs, and PC axon collaterals [27, 60–64]. It is possible that a combination of multiple mechanisms could corroborate to generate the synchronization of PC firing [64].

One of the main contributions to the synchronization of PC simple spikes is the excitation input from GCs [45, 61]. GC synchronization is a general phenomenon that could be induced by Golgi cells [65, 66]. Here we delivered a burst protocol mimicking the effect of GC

synchronized spikes [42] and investigated how they are propagated to PCs. We then studied how the precise PC synchronization is affected by the STP of GC-PC synapses and strong inhibition from MLIs. We found that GC bursts can trigger PCs to generate a significant level of synchronization, mainly due to the slow spillover component of GC-PC synapses that provides persisting excitation inputs. As a result, excitation slowly returns to baseline, and there is enough time to accumulate excitation continually approaching the spike threshold. For instance, with a 300 Hz burst, the interspike interval is only about 3 ms that makes the slow spillover component staying at a high level to easily produce a spike for the next incoming stimulus.

In more realistic scenarios, excitation and inhibition are entangled. Our results here suggest that PC synchronization could result from different effects of excitatory STP and MLI inhibition on precise temporal dynamics of simple spikes, which then are modulated by the balance of excitation and inhibition, together with input burst spike patterns, such as the duration and number of spikes in a burst.

### Pause response in PC firing

Another unique feature of temporal spiking pattern is the pause response shown here for PCs. A delayed and adaptively timed pause in PCs simple spike firing is thought to play an essential role in the computation performed by PCs [67–70], as well as modulate downstream deep cerebellar nucleus neurons [71]. This type of response has been ascribed to intrinsic mechanisms and long-term depression of parallel fibers converged to PC synapses [43, 70, 72–75].

Here, we investigated how pause response is elicited by GC-PC and GC-MLI-PC neural pathways. The pause is more likely to appear when both neural pathways are incorporated, which is presumably due to that the offset of inhibition is regulated by short-term dynamics of excitability. Burst input can induce a significant pause that is enforced by MLIs. Overall, all the four parameters of MLI-PC inhibition, the number of synapse, number of MLI-MLI recurrent connection, synaptic weight, and synaptic short-term dynamics, could contribute, in different ways, to the change of MLI-PC synaptic transmission. However, consistent with the previous findings [74], the main factor is the enhancement of MLI-PC synaptic weight that increases the duration of pause response and provides evidence that spike pauses are mainly regulated by MLI. We also observed that MLI-PC and MLI-MLI inputs regulate the pause in an opposite fashion, and the input frequency has no significant effect on pause, while the pause duration increase as the inhibition strength increases, which is consistent with experimental results [76].

### Limitations

Here we explored how PC dynamics can be modulated by synaptic inputs from two neural pathways. It is also known that neuronal dynamics can be modulated by various intrinsic properties. One of which is neuronal morphology. Especially PCs have a complex morphological structure in which parallel fibers and climbing fibers are targeted at different parts of the dendritic tree [8, 77]. Although the complex dendritic tree can be reduced to a simple structure while preserving basic characteristics of firing activity [78], the PC dynamics is suggested to branch-specific due to the distribution of ion channels [77]. These intrinsic mechanisms can dramatically regulate PC dynamics. For instance, $Ca^{2+}$-activated $K^+$ channels behave like a high-pass filter that allows PC to respond GC high frequency inputs [79]. For MLI inhibitory synaptic inputs on PCs, it is also suggested that the input from stellate cells is often located at spiny dendrites, whereas basket cells are more on the soma and smooth dendrites [47], which induce different inhibition strengths to PCs. Here we mimicked a wide range of inhibition strengths by varying the strength of MLI-PC synapses. However, these explicit components, dendritic locations of synapses and ion channels, are not explored in this study.

In the cerebellar granular layer, Golgi cells provide inhibition input for GCs and form both feedforward and feedback inhibition loops to regulate GC dynamics [80]. It has been suggested that GC network dynamics of synchronization and oscillation are related to the feedback loop between GoC and GC [65, 66, 81], and gap junctions between GoCs that can promote synchronous oscillatory patterns [26]. Therefore, Golgi cells could contribute to the modulation of PC dynamics [80, 82, 83]. Further studies are needed to explore the role of Golgi cells, together with GCs and MLIs, in regulating PC dynamics.

Here we only considered the dynamics of PC simple spikes. Another unique feature is that PCs also receive strong excitation from climbing fibers represented by distinctive responses known as complex spikes, which play an important role in cerebellar learning [77]. Moreover, climbing fibers also regulate the dynamics of MLIs via synaptic spillover [84] and neighbor PCs via ephaptic coupling [85]. Therefore, they could potentially contribute to the PC network dynamics [13]. Further work is needed to elucidate these unaddressed questions.

## Supporting information

**S1 Fig. Related to Fig 1. The variability of PC responses in the network induced by the randomization of parameter setting in the model.** (A) Membrane potential traces of three example PCs triggered by 20 Hz Poisson spikes. (B) The corresponding ISI (interspike interval) distribution. (C) The input-output relationship represented by PC firing a function of input GC spikes. Here the simulation is conducted in the baseline condition, where there is no inhibitory MLI (MLI off) and no excitatory STP on GC-PC synapses (STP off, *i.e.*, synaptic dynamics is not subjected to the modulation of short term plasticity).
(TIF)

**S2 Fig. Related to Fig 1. The dynamics of short-term plasticity (STP) with different settings of parameters and stimulation protocols.** (A) The STP dynamics with different parameters of time constants: $\tau_{fac}$ for facilitation and $\tau_{rec}$ for depression, under two settings of initial efficacy U for GC-PC synapses and MLI-PI synapses. (Left) EPSPs triggered by a train of 10 spike at 200 Hz at $U_{exc} = 0.06$ for facilitation and $U_{exc} = 0.42$ for depression, with different facilitation time constants ($\tau_{fac}$) and recovery time constants ($\tau_{rec}$). (Right) Similar to EPSPs but for IPSPs with $U_{inh}$. (B) STP described by the ratio $PSP_n/PSP_1$ (EPSPs, top; IPSPs, bottom) showing facilitation or depression triggered by a train of different burst spikes at 50, 100 and 300 Hz with a wide range of U values (0.02-1).
(TIF)

**S3 Fig. Related to Fig 2. Nonlinear gain control of the PC I-O curves with different settings.** (A) The profiles of PC firing with an opposite (in contrast to the default values) pair of STP initial efficacy U values: $U_{exc} = 0.08$ for GC-PC facilitation and $U_{exc} = 0.02$ for MLI-PC depression, under different conditions. Here STP refers to GC-PC STP, and MLI refers to MLI-PC inhibition. Baseline without STP and MLI (-STP-MLI); STP ON without MLI (+STP-MLI); STP off with MLI but no MLI-PC STP (-STP+MLI(-STP)); STP off with MLI and MLI-PC STP (-STP+MLI(+STP)); STP on with MLI but no MLI-PC STP (+STP+MLI(-STP)); STP on with MLI and MLI-PC STP (+STP+MLI(+STP)). Each point is mean±SD (n = 50). (B) Similar to A but GC-PC synapses are depressing and MLI-PC synapses are facilitating. Here the STP of MLI-PC synapses is switched off, compared to Fig 2. $W_{MLI-PC} = 3.5$ nS gives strong inhibition and depresses PC firing (left). The I-O profiles are more visible with $W_{MLI-PC} = 1.2$ nS. Lines are fits to a Hill function.
(TIF)

**S4 Fig. Related to [Fig 3](). The PC I-O curves affected by GC-PC STP and MLI inhibition in different ways.** (A) The total excitation input $G_{exc}$ as a function of GC input at various levels of synaptic efficacy $U_{exc}$ (0.01-0.09) without short-term facilitation (STF) (left) and with STF (right). (B) The gain change without and with STP for different levels of inhibition (left), and without and with inhibition for different levels of STP $U_{exc}$ (right), compared to the baseline for each case. The MLI inhibition weights are changed by a scaling factor as shown in the axis index. The default parameter values are used here, such that +STP means $U_{exc} = 0.4$, and +MLI means $W_{MLI} = 3.5$ nS.
(TIF)

**S5 Fig. Related to [Fig 3](). The I-O function of each individual component of GC-PC synapses.** (A) Excitation $G_{exc}$ of AMPA slow component (left) and AMPA fast component (right) as a function of GC input at various levels of synaptic efficacy $U_{exc}$ (0.1-0.75) without short-term depression (STD) (top) and with STD (bottom). (B) Similar to (A) but for lower values of $U_{exc}$ (0.01-0.09) without short-term facilitation (STF) (top) and with STF (bottom).
(TIF)

**S6 Fig. Related to [Fig 7](). Synchronization and pause in the PC network under different settings.** (A) The PC network shows similar behaviors with (blue) and without (red) MLI-PC STP. (B) Similar to [Fig 7](), but with an opposite pair of U values: $U_{exc} = 0.08$ for GC-PC facilitation and $U_{exc} = 0.2$ for MLI-PC depression, in contrast to [Fig 7](). Without short-term depression in MLI-PC synapses, the inhibition is too strong to suppress PC firing, which indicates the need of a balance of excitation and inhibiting, as also revealed by Figs [9](–[11]().
(TIF)

**S7 Fig. Related to [Fig 2]() and [S6 Fig](). Time courses of the excitatory conductance and short-term variable.** Time courses of PC population firing rate (red), total excitatory conductance averaged over all PCs ($G_{exc}$, green) and raster plots of excitatory conductance of each PC, and short-term plasticity R variable averaged over all PCs (blue) and raster plots of R of each PC. The same pairs of U values under different settings as in [S6 Fig]() GC-PC synapses are depressed due to STP, and R variable reduced to be close to 0 during burst inputs.
(TIF)

**S8 Fig. Related to [Fig 8](). PC firing rate under different burst frequencies.** PC population firing rate under bursts of 10 spikes at 100, 200, and 300 Hz in four different conditions, baseline, with MLI, with STP, and with both MLI and STP. The background Poisson stimulation frequency is 20 Hz. The PC population firing rate tends to be sustained in the baseline. Adding MLI tends to build up dynamics over stimulus, in particular for high frequencies. Adding STP tends to make dynamics transient and decay over stimulus. Using MLI and STP together makes the dynamics more transient, so that the first peak is more prominent.
(TIF)

**S9 Fig. Related to [Fig 8](). PC firing rate under different burst spikes.** PC population firing rate under bursts of 200 Hz with 2, 5, and 7 spikes in four conditions. The background Poisson stimulation is 20 Hz.
(TIF)

## Author Contributions

**Conceptualization:** Lingling An, Jian K. Liu.

**Data curation:** Yuanhong Tang, Jian K. Liu.

**Formal analysis:** Yuanhong Tang, Jian K. Liu.

**Funding acquisition:** Lingling An, Qingqi Pei, Quan Wang, Jian K. Liu.

**Investigation:** Yuanhong Tang, Lingling An, Jian K. Liu.

**Methodology:** Yuanhong Tang, Lingling An, Jian K. Liu.

**Project administration:** Lingling An, Jian K. Liu.

**Resources:** Lingling An, Jian K. Liu.

**Software:** Yuanhong Tang, Ye Yuan, Jian K. Liu.

**Supervision:** Lingling An, Quan Wang, Jian K. Liu.

**Validation:** Yuanhong Tang, Jian K. Liu.

**Visualization:** Yuanhong Tang, Ye Yuan, Jian K. Liu.

**Writing – original draft:** Yuanhong Tang, Lingling An, Jian K. Liu.

**Writing – review & editing:** Yuanhong Tang, Lingling An, Jian K. Liu.

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
