## [Decision Letter · Decision Letter 0]

4 Sep 2020

Dear Dr An,

Thank you very much for submitting your manuscript "Modulation of neural dynamics through the interaction of excitatory and inhibitory feedforward pathways" for consideration at PLOS Computational Biology.

As with all papers reviewed by the journal, your manuscript was reviewed by members of the editorial board and by several independent reviewers. In light of the reviews (below this email), we would like to invite the resubmission of a significantly-revised version that takes into account the reviewers' comments. 

From reading the reports, all the points raised seem reasonable and should be addressed in a reworked manuscript. In particular, the reviewers were concerned about clarity of the main message, as well as model choices. A revised manuscript should have substantial effort devoted to robustness tests and/or motivation of the parameters.

We cannot make any decision about publication until we have seen the revised manuscript and your response to the reviewers' comments. Your revised manuscript is also likely to be sent to reviewers for further evaluation.

Sincerely,

Alex Cayco Gajic

Guest Editor

PLOS Computational Biology

Lyle Graham

Deputy Editor

PLOS Computational Biology

Reviewer's Responses to Questions

**Comments to the Authors:**

Reviewer #1: Uploaded as attachment.

Reviewer #2: In the cerebellum, Purkinje cells (PCs) receive feedforward inputs mainly from two neural pathways: the first one is the feedforward excitatory pathway of parallel fibers from granule cells (GCs) to PCs; the second one is the feedforward inhibition pathway from GCs, via molecular layer interneurons (MLIs), to PCs. This manscuript [PCOMPBIOL-D-20-00987] have systematically invesigatted how the dynamics of Purkinje cells (PCs) are jointly modulated by these two feedforward pathways. To this end, the authors established a simplified PC network with short-term plasticity, in which the firing dynamics of GCs, PCs and MLIs are simulated with the integrate-and-fire neurons, and different stimulation protocols of mossy fiber inputs are used to drive the network activity. Briefly, the authors showed that the interaction of feedforward pathways of excitation and inhibition, together with synaptic short-term dynamics, can dramatically affect the PC responses that consequently change the network dynamics of the PC circuit. The topic of this manuscript is quite interesting, and the manuscript is well organized and written. I only have a few comments and suggestions that the authors may consider during revision.

Comments and suggestions:

1. Although the topic of this study is interesting, the title of this manuscript is inappropriate. Indeed, it does not contain any word related to the cerebellum, making the title seems to be too general. Please revised to clearly explain the research.

2. I strongly suggest the authors focus on the PC network in this study, and several general sentences should be removed from the manuscript, in particular in the abstract and introduction.

3. In Fig 3, the authors showed that GC-PC synaptic STP with the aid of MLI inhibition can significantly enhance gain modulation of PC firing dynamics. This is one key finding of this manuscript because gain control of cortical neurons are believed to be highly associated with their neuronal computations. I suggest the authors to compare their modeling findings with experimental results on PCs, or at the very least I would expect you to further discuss it.

4. In this study, both the conduction delay on fibers and the synaptic transmission delay are not considered in the PC network. In fact, transmission delay might play important roles in controlling the dynamics of the PC networks. In particular, it might significantly impact the PC firing phase and contribute to the phase shift in neural dynamics in response to time-varying inputs.

5. In the cerebellar granular layer, there are several Golgi cells. The GoCs inhibit the numerous GCs through sagittal branching of their axons and, in addition, they are also connected together by gap junctions. Recent modeling studies have shown that GoCs may play roles in the emergence of granular oscillations. Although the authors did not include the Golgi cells in their PC network model, I still suggest them to further discuss the possible regulations of GoCs in the dynamics of PCs co-modulated by feedforward GC-PC and MLI-PC pathways.

6. There are too many references in the manuscript but several important references are missing. The authors may be interested in the following recent published papers: SK Sudhakar et al., PLoS Comput. Biol., 2014 & 2015; Y Zang & E De Schutter, Front. Syst. Neurosci., 2019.

Reviewer #3: This paper employs a a basic model of a cerebellar circuit to address how the interaction of the feed-forward inhibitory pathway and the short term plasticity of the feed-forward excitatory pathway from granule cells control Purkinje cell output and associated circuit activity.

While I consider that this is an important subject in the context of cerebellar circuit dynamics, the paper fails to deliver a clear message and convincing results regarding the generality of the model beyond the specific choice of description level and parameters.

Here are a few suggestions to improve the paper:

1-Authors could emphasize from the beginning the model description level: point neurons implemented with an integrate and fire model. Nothing wrong about this choice, but of course limitations arise from how the intrinsic neuron dynamics integrates the combined effect of the two different pathways. Authors could further discuss how a morphological distribution of the inputs and more complex active channel description in PC cells could build up in the mechanisms described. Limitations and generality of the results with respect to the chosen approach should be discussed in further detail.

2-Beyond noise, authors do not provide a validation of the robustness of the described phenomena as a function of the parameters of the neuron model, just the dependence of the dynamic synapse parameters is assessed. This can be easily addressed in more simplified models as the one used in this study. Furthermore, no justification of the parameters chosen is provided, in particular time constants.

3-The role of climbing fibers in only hinted in the discussion. it should at least be introduced at the beginning of the manuscript and a justification of why they are left out could be provided.

Minor issues:

1-Please clarify what stimulation at -60, 70, 0 mV is in Fig. 1B

2-Line 128, please justify the choice of the model from Ref. 20.

3-Please justify the choice of all parameters in section IV and table I.

4-Please fix discrepancies between labels and figure caption in Fig. S7

**Have all data underlying the figures and results presented in the manuscript been provided?**

Reviewer #1: Yes

Reviewer #2: Yes

Reviewer #3: **No: **No link to the model code or data is provided

PLOS authors have the option to publish the peer review history of their article (what does this mean?). If published, this will include your full peer review and any attached files.

Reviewer #1: No

Reviewer #2: No

Reviewer #3: No
---

## [Decision Letter · Decision Letter 1]

23 Nov 2020

Dear Dr An,

Thank you very much for submitting your manuscript "Modulation of the dynamics of cerebellar Purkinje cells through the interaction of excitatory and inhibitory feedforward pathways" for consideration at PLOS Computational Biology. As with all papers reviewed by the journal, your manuscript was reviewed by members of the editorial board and by several independent reviewers. 

The reviewers found the manuscript to have improved, however there still remain some concerned outlined by Reviewer 1. These points seem reasonable and should be addressed in your next submission.

Sincerely,

Alex Cayco Gajic

Guest Editor

PLOS Computational Biology

Lyle Graham

Deputy Editor

PLOS Computational Biology

[LINK]

The reviewers found the manuscript to have improved, however there still remain some concerned outlined by Reviewer 1. These points seem reasonable and should be addressed in your next submision.

Reviewer's Responses to Questions

**Comments to the Authors:**

Reviewer #1: The manuscript “Modulation of the dynamics of cerebellar Purkinje cells through the interaction of excitatory and inhibitory feedforward pathways” uses reduced network modelling of the cerebellar cortex to study how the interaction of excitatory and inhibitory feedforward pathways affect response of cerebellar Purkinje cells. The reviewer acknowledges the substantial efforts of the authors to include further simulations, analyses and parameter exploration, and the consistency of these results with previous findings strengthen the paper. However, there are some questions raised by the apparent robustness of the observations to the precise form of STP, that would need to be discussed, as well as some suggestions on improving the clarity of the text and methods.

Major comments:

1) Figure 2 characterizes the change in gain of PC Input-Output function, as the change in average slope. However, the I-O functions are highly non-linear (with a saturating non-linearity). The gain calculation in reference [28] fit a convolution of Hill function and exponential to the measured I-O. Can the authors please clarify (in methods and main text) if they also fit a similar non-linear function, and refer to the slope parameter in that model, and not to a linear fit to the simulation data?

Further, in Figure 2B and 2C, can the legend clarify what the lines and scatter are (fits versus simulation?) Also, to support the statement that “MLI-mediated inhibition introduces an additive shift” on line 166-167, it seems necessary to additionally show changes in offset, along with gain in Figure 2D?

2) For Supplementary Figure 3A, the black and cyan lines (without STP at MLI-PC synapses) seem to have no PC output – is that because of inappropriate scaling of inhibitory conductances? Can that be recovered by reducing the weight of MLI-PC synapses?

It is also difficult to compare the results in S3A and S3B directly because fixing U (in no STP cases) at different values also changes scaling of synaptic currents? If the claim is that the results are robust to those parameters, perhaps the quantification of gain/offset change can be added to the figure, or highlighted as the corresponding column in Figure S4B? Further, can the authors discuss (in addition to line 174-176) why the two curves end up looking similar in the two scenarios – is it mainly that for all values of U, synapses become depressing at high enough stimulation frequency (resource variable R goes near 0)? This suggests that there is still a difference at lower frequencies – of facilitation versus depression – which can be seen in Figure 3B?

3) Line 390-399: Cited reference [51] is mentioned as showing linear IO response of PCs without MLI-mediated inhibition (which is not entirely consistent with the results perhaps, Fig3A for example). Further, it is unclear how this links to the effect of turning off STP at GC-PC synapses (line 391-392). Is it mentioned as an alternative mechanism for generating nonlinear IO responses? The experimental recordings did not eliminate STP and still observed linear response curves, which is again not entirely consistent with these modelling results. Although the photolysis of glutamate experiment do bypass synaptic dynamics, the electrical stimulation of parallel fibres also produced linear PC synapses. As these stimulations are not necessarily linearly related to input frequency/strength, it may be important to mention this caveat to explain the discrepancy between the data and model, or to exclude this reference/discussion altogether.

Minor comments:

1) As was mentioned in the previous round, the default parameters used in this study for Uexc at GC-PC and Uinh for MLI-PC differ from published experimental results and previous modelling (see refs: 18, 19, 27, 36). In particular, in figure 1B, although kinetics of single EPSP are fitted, as well as STP for GC-MLI synapses, the STP dynamics for GC-PC synapses, (and MLI-PC), which are the main objects of study here, are specifically not fitted. Although some results are robust to the parameter choice (partly because all synapses become depressing at high frequency stimulation), it may be better to explicitly mention that other aspects of PC response may still vary for different STP dynamics.

2) For Figure S4B, can the legend clarify whether the change in gain was compared to a single fixed IO curve, or to the corresponding baseline for each case (for example for each Uexc, it is compare to an IO with the same U with and without its dynamics?)

3) Line 129: “ a single MLI of inhibition” -> “inhibition from a single MLI”

4) Line 193-194 about gain control being maintained is unclear. Do the authors mean the change in excitatory conductance with stimulation frequency is similar across different values of Uexc (however there is a large difference between them for 10-50Hz inputs, and reduced difference for 60-140Hz inputs).

5) Line 252-255: The cited reference [35] shows that regularity of spiking is not an important feature for the fast “rate” coding by Purkinje cells. Thus it may be inappropriate to cite that to suggest that differences in ISI and CV (dependent on STP and inhibition) change information transmission by PCs, as the claim in the reference is opposite.

6) Figure 7 and S6: Although the pause response doesn’t depend on precise U, is it due to the specific stimulation frequency (i.e. at 100Hz – 300Hz), all GC-PC synapses become effectively depressed? Plotting the evolution of Gexc, R variables may be insightful.

7) Line 285: Consisting -> Consistent

8) Line 287-288: It is unclear what the main conclusion of Fig 8A and FigS7 is. Can the authors help the reader focus on what is meant by “different PC dynamics”?

9) Line 409: depression -> depressing; Line 410: facilitation->facilitating

Reviewer #2: This manuscript has been significantly improved during revision. I therefore suggest the publication of this study in its current version.

Reviewer #3: Thank you for addressing my concerns. I think the new version of the paper is now much stronger.

**Have all data underlying the figures and results presented in the manuscript been provided?**

Reviewer #1: Yes

Reviewer #2: Yes

Reviewer #3: **No: **The authors mention in the response to the reviews that "Our model with the code will be released to the public as part of open science." but no link is available in the manuscript.

PLOS authors have the option to publish the peer review history of their article (what does this mean?). If published, this will include your full peer review and any attached files.

Reviewer #1: No

Reviewer #2: No

Reviewer #3: No
---

## [Editor Report · Decision Letter 2]

4 Jan 2021

Dear Dr An,

We are pleased to inform you that your manuscript 'Modulation of the dynamics of cerebellar Purkinje cells through the interaction of excitatory and inhibitory feedforward pathways' has been provisionally accepted for publication in PLOS Computational Biology.

Best regards,

Alex Cayco Gajic

Guest Editor

PLOS Computational Biology

Lyle Graham

Deputy Editor

PLOS Computational Biology

---

## [Editor Report · Acceptance letter]

5 Feb 2021

PCOMPBIOL-D-20-00987R2 

Modulation of the dynamics of cerebellar Purkinje cells through the interaction of excitatory and inhibitory feedforward pathways

Dear Dr An,

I am pleased to inform you that your manuscript has been formally accepted for publication in PLOS Computational Biology. Your manuscript is now with our production department and you will be notified of the publication date in due course.

With kind regards,

Alice Ellingham
